# Molecular tracking devices quantify antigen distribution and archiving in the murine lymph node

Shannon M Walsh[1†], Ryan M Sheridan[2†], Erin D Lucas[3,4†], Thu A Doan[3,5],
Brian C Ware[3,4], Johnathon Schafer[5], Rui Fu[2], Matthew A Burchill[5],
Jay R Hesselberth[1,2], Beth Ann Jiron Tamburini[3,4,5]*

[1]Department of Biochemistry and Molecular Genetics, University of Colorado School of Medicine, Aurora, United States; [2]RNA Bioscience Initiative, University of Colorado School of Medicine, Aurora, United States; [3]Immunology Graduate Program, University of Colorado School of Medicine, Aurora, United States; [4]Department of Immunology and Microbiology, University of Colorado School of Medicine, Aurora, United States; [5]Department of Medicine, Division of Gastroenterology and Hepatology, University of Colorado School of Medicine, Aurora, United States

**Abstract** The detection of foreign antigens in vivo has relied on fluorescent conjugation or indirect read-outs such as antigen presentation. In our studies, we found that these widely used techniques had several technical limitations that have precluded a complete picture of antigen trafficking or retention across lymph node cell types. To address these limitations, we developed a 'molecular tracking device' to follow the distribution, acquisition, and retention of antigen in the lymph node. Utilizing an antigen conjugated to a nuclease-resistant DNA tag, acting as a combined antigen-adjuvant conjugate, and single-cell mRNA sequencing, we quantified antigen abundance in the lymph node. Variable antigen levels enabled the identification of caveolar endocytosis as a mechanism of antigen acquisition or retention in lymphatic endothelial cells. Thus, these molecular tracking devices enable new approaches to study dynamic tissue dissemination of antigen-adjuvant conjugates and identify new mechanisms of antigen acquisition and retention at cellular resolution in vivo.

*For correspondence:
beth.tamburini@cuanschutz.edu

†These authors contributed equally to this work

Competing interests: The authors declare that no competing interests exist.

## Introduction

Depending on the route of infection, vaccination mode, and ability of antigens to traffic, different dendritic cell (DC) subsets are required to initiate T cell priming. Upon subcutaneous immunization, small soluble proteins and virus particles pass through the lymphatics to the lymph node (LN), where LN-resident DCs acquire and present antigen (*Manolova et al., 2008*; *Gerner et al., 2017*). For larger antigens and/or pathogens that are too large to pass through the lymphatic capillaries, dermal DCs migrate to the LN for presentation of processed antigens to naive T cells (*Manolova et al., 2008*; *Bonneau et al., 2006*; *Hampton and Chtanova, 2019*). Most adaptive immune responses require antigen processing and presentation by conventional DCs in either the draining LN or at the site of infection or vaccination (migratory cutaneous or dermal DCs) (*Eisenbarth, 2019*).

Previous studies have shown that viral antigens persist in the LN beyond the time frame of infectious virus (*Jelley-Gibbs et al., 2005*; *Kim et al., 2010*; *Kim et al., 2011*; *Takamura et al., 2010*; *Woodland and Kohlmeier, 2009*; *Zammit et al., 2006*). We recently found that lymphatic endothelial cells (LEC) store antigens from viral infection and vaccination (*Kedl et al., 2017*; *Kedl and Tamburini, 2015*; *Tamburini et al., 2014*). Using a vaccine formulation that elicits robust cell-mediated

**eLife digest** The lymphatic system is a network of ducts that transports fluid, proteins, and immune cells from different organs around the body. Lymph nodes provide pit stops at hundreds of points along this network where immune cells reside, and lymph fluid can be filtered and cleaned. When pathogens, such as viruses or bacteria, enter the body during an infection, fragments of their proteins can get swept into the lymph nodes. These pathogenic proteins or protein fragments activate resident immune cells and kickstart the immune response. Vaccines are designed to mimic this process by introducing isolated pathogenic proteins in a controlled way to stimulate similar immune reactions in lymph nodes.

Once an infection has been cleared by the immune system, or a vaccination has triggered the immune system, most pathogenic proteins get cleared away. However, a small number of pathogenic proteins remain in the lymph nodes to enable immune cells to respond more strongly and quickly the next time they see the same pathogen. Yet it is largely unclear how much protein remains for training and how or where it is all stored. Current techniques are not sensitive or long-lived enough to accurately detect and track these small protein deposits over time.

Walsh, Sheridan, Lucas, et al. have addressed this problem by developing biological tags that can be attached to the pathogenic proteins so they can be traced. These tags were designed so the body cannot easily break them down, helping them last as long as the proteins they are attached to. Walsh, Sheridan, Lucas et al. tested whether vaccinating mice with the tagged proteins allowed the proteins to be tracked. The method they used was designed to identify individual cell types based on their genetic information along with the tag. This allowed them to accurately map the complex network of cells involved in storing and retrieving archived protein fragments, as well as those involved in training new immune cells to recognize them.

These results provide important insights into the protein archiving system that is involved in enhancing immune memory. This may help guide the development of new vaccination strategies that can manipulate how proteins are archived to establish more durable immune protection. The biological tags developed could also be used to track therapeutic proteins, allowing scientists to determine how long cancer drugs, antibody therapies or COVID19 anti-viral agents remain in the body. This information could then be used by doctors to plan specific and personalized treatment timetables for patients.

immunity comprising antigen, a Toll-like receptor (TLR) agonist, and an agonistic αCD40 antibody (TLR/αCD40 vaccination) or viral infection (*Ahonen et al., 2004*; *Ahonen et al., 2008*; *Badovinac et al., 2002*; *Corbin and Harty, 2004*; *Kaech and Ahmed, 2001*; *Kurche et al., 2010*; *Kurche et al., 2012*; *Lucas et al., 2018*; *McWilliams et al., 2010*; *Sanchez and Kedl, 2012*; *Sanchez et al., 2007*; *Tamburini et al., 2012*), we discovered that antigens were durably retained in the LN (*Kedl et al., 2017*; *Kedl and Tamburini, 2015*; *Tamburini et al., 2014*). Antigen storage was dependent on the presence of a TLR agonist (e.g. polyI:C alone [TLR3/MDA5/RIGI or Pam3cys (TLR1/2)+ αCD40]), but also occurred with antigen conjugated to a TLR agonist (e.g. 3M019 [TLR7]) [*Tamburini et al., 2014*]. We named this process 'antigen archiving' and showed it is important to poise memory T cells for future antigenic encounters (*Tamburini et al., 2014*).

Prior to these studies, the only non-hematopoietic cell type thought to retain antigens were follicular DCs, which harbor antigens in antigen-antibody complexes for extended periods of time and for the benefit of B cell memory (*Zammit et al., 2006*; *Heesters et al., 2013*). Fibroblasts and non-endothelial stromal cells (SCs) comprise a large portion of the LN stroma and are capable of presenting peripheral tissue antigens, but their capacity to acquire and present foreign antigens is not yet well understood (*Fletcher et al., 2010*; *Fletcher et al., 2011*; *Turley et al., 2010*). We were unable to detect antigen archiving by blood endothelial cells (BECs) or fibroblasts in our initial studies (*Kedl et al., 2017*; *Kedl and Tamburini, 2015*). While LECs have been shown to present antigens in the absence of inflammation to induce T cell tolerance (*Cohen et al., 2010*; *Cohen et al., 2014*; *Nichols et al., 2007*; *Rouhani et al., 2015*; *Tewalt et al., 2012*; *Dubrot et al., 2014*; *Hirosue et al., 2014*; *Lund et al., 2012*), we showed that presentation of archived antigen occurs only after exchange of the archived antigen from an LEC to a migratory DC; changing the stimulus from

tolerizing to immunostimulatory (*Kedl et al., 2017*; *Kedl and Tamburini, 2015*). Soluble antigens are exchanged via two distinct mechanisms: (i) direct exchange between LECs and migratory DCs and (ii) LEC death. Antigen transfer from LECs to both migratory conventional (c)DC1s and cDC2s is required for archived antigen presentation to antigen-specific memory T cells (*Kedl et al., 2017*; *Kedl and Tamburini, 2015*). After viral infection, archived antigen is transferred to *Batf3*-dependent migratory DCs as a result of LEC death during LN contraction (*Kedl et al., 2017*).

Limitations of current approaches have precluded sensitive and quantitative measures of antigen levels across cell types, providing only a glimpse of the cell types and molecular mechanisms that control antigen acquisition, processing, and retention in the LN. Studies of antigen in the LN and peripheral tissues have mainly relied on antigen-fluorophore conjugates or indirect measurement of antigen uptake and presentation (*Gerner et al., 2017*; *Jelley-Gibbs et al., 2005*; *Kim et al., 2010*; *Zammit et al., 2006*; *Kedl et al., 2017*; *Tamburini et al., 2014*; *Jelley-Gibbs et al., 2007*), which defined antigen acquisition by specific DC subsets and trafficking of antigens using live imaging (*Gerner et al., 2017*). However, antigen archiving has been difficult to study because antigen-fluorophore conjugates suffer from low microscopic detection sensitivity, yielding weak signals that diminish over time. Moreover, detection of antigen in the LN and other tissues has relied on flow cytometric analysis using cell surface markers, restricting analysis to specific cell types. To address these limitations and better understand antigen archiving, we developed a new approach to track an antigen-phosphorothioate DNA. The phosphorothioate DNA contained a tracking device for detection using single-cell mRNA sequencing and initiated a robust immune response when conjugated to the protein antigen. Here, we outline the tissue distribution in vivo of this antigen-DNA conjugate by utilizing the conjugated phosphorothioate DNA as an adjuvant and tracking device.

## Results

### Generation, validation, and immunogenicity of antigen-DNA conjugates

To quantify the dissemination and uptake of antigen in the draining LN after vaccination, we developed a vaccination strategy to measure antigen levels using single-cell mRNA sequencing. Many prior studies have used the model antigen, ovalbumin (ova), conjugated to a fluorophore to track antigen in vivo. Here, we conjugated ova to DNA oligonucleotides with barcodes suitable for analysis by single-cell mRNA sequencing (*Figure 1a*). The ~60 nt DNA tag contains a unique sequence barcode and PCR primer binding sites, similar to CITE-seq tags (*Stoeckius et al., 2017*; *Figure 1— source data 1*). We measured the stability of unconjugated DNA and ova-DNA conjugates in which the conjugated DNA either had normal phosphodiester linkages (pDNA) or was protected throughout by phosphorothioate linkages (psDNA). Quality control of these conjugates indicated a 1:1 stoichiometry of protein to DNA (*Figure 1b*). To measure the stability of the antigen-DNA conjugate, we added antigen-DNA conjugates to cultures of bone marrow-derived dendritic cells (BMDCs) and quantified the amount of DNA in cell lysates and media over time using the PCR handle to detect the DNA by quantitative PCR. Amount of DNA was quantified as a ratio of DNA detected relative to the amount of protein acquired from the cell lysate. We found significantly higher levels of ova-psDNA in cells relative to ova-pDNA (approximately fourfold at day 1; p=0.002 and approximately sevenfold at day 3; p=0.004), indicating that psDNA is more stable than pDNA (*Figure 1c*). In addition, ova conjugation was required for phagocytosis by BMDCs as we detected limited amounts of unconjugated pDNA or psDNA (values <1 at days 1–7) (*Figure 1c*). To determine if the BMDCs had both the ova and DNA within each cell, we used flow cytometry and immunofluorescence using an antibody to detect ova and streptavidin to detect the biotinylated DNA tag. We detected both ova and DNA within the same cells by flow cytometry (*Figure 1d*, *Figure 1—figure supplement 1a*) and co-localization by immunofluorescence (*Figure 1e*). We also measured conjugate stability in mouse LECs, a cell type that retains foreign proteins for long periods (*Tamburini et al., 2014*), and found that ova-psDNA conjugates were stable over 7 days of culture, whereas ova-pDNA was rapidly degraded (*Figure 1f*). In the endothelial cells, we detected both the ova protein and the barcode within the same cell and co-localized to same location (*Figure 1—figure supplement 1b, c*). Furthermore, the ova-psDNA retention within the LECs was similar to a vaccine strategy using an ova protein-fluorophore conjugate with polyI:C and anti-CD40, which we previously demonstrated induces antigen archiving (*Tamburini et al., 2014*; *Figure 1—figure*

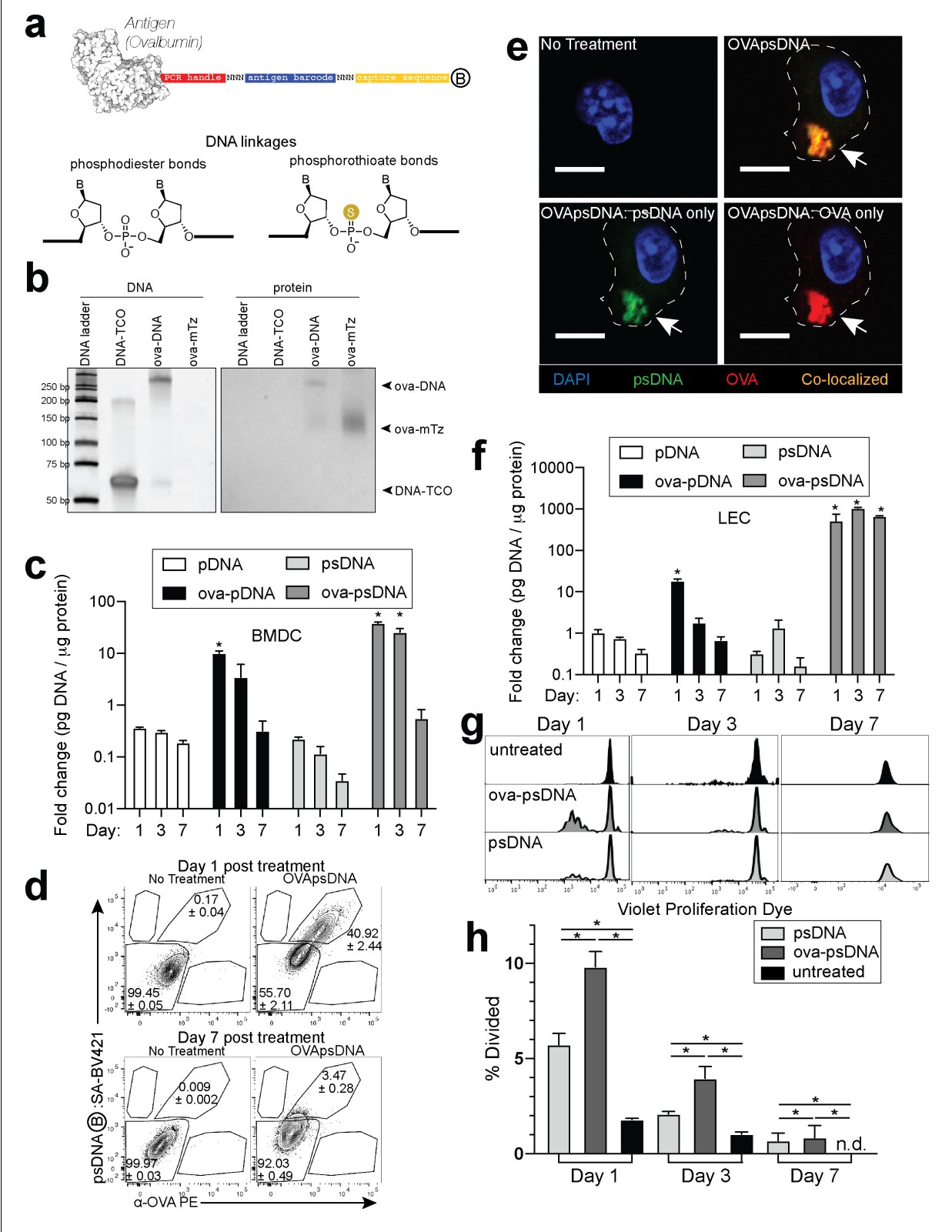

**Figure 1.** Antigen-psDNA conjugates undergo normal processing and presentation. (**a**) Schematic of ovalbumin (PDB code 1ova) antigen conjugation to barcoded DNA with phosphodiester and phosphorothioate DNA linkages and a 3′ biotin label (circle with B inside). Sulfur replaces a non-bridging oxygen to create a DNA phosphorothioate linkage. List of oligo sequences used can be found in *Figure 1—source data 1*. (**b**) Conjugation of oligonucleotides to ovalbumin. Purified conjugate was analyzed by 10% TBE native PAGE stained with GelRed for DNA (left) followed by Coomassie

*Figure 1 continued on next page*

*Figure 1 continued*

staining for protein (right). DNA-TCO: 61 nt barcoded oligonucleotide with 5'-trans-cyclooctene (TCO); ova-mTZ: ovalbumin functionalized with methyltetrazine (mTZ); ova-DNA: DNA-conjugated ovalbumin product with oligonucleotide attached. (c) Bone marrow-derived dendritic cells (BMDCs) were treated with pDNA, psDNA, ova-pDNA, or ova-psDNA (5 μg) by addition to the culture media. After 1, 3, and 7 days, cells were washed, released, lysed, and analyzed for pDNA or psDNA by qPCR. Values are displayed as fold-change relative to the negative control (cells alone). Asterisks denote sample significant amounts relative to the negative control (p<0.01; Wilcoxon rank-sum test). Error bars represent standard error of the mean (SEM). 3–5 wells were evaluated per group on 2–3 independent occasions. (d) Flow cytometric analysis of ova-psDNA conjugates acquired by BMDCs after 1 day or 7 days. Cells were washed 1 day after ova-psDNA treatment. Harvested BMDCs were stained with anti-ovalbumin made in rabbit and a secondary anti-rabbit conjugated to Phycoerythrin (PE) and then stained with streptavidin conjugated to brilliant violet 421 to visualize the 3' biotin label on the psDNA. Shown are average and ± standard error. Experiment was performed three times with three technical replicates. (e) As in (d) except cells were plated onto glass coverslips and treated with ova-psDNA for 24 hr prior to staining with either anti-ovalbumin and a secondary conjugated to PE (red) followed by streptavidin conjugated to Fluorescein Isothiocyanate (FITC) (green). Co-localization is shown in yellow. Scale bar is 10 μm. Imaging was repeated three independent times. Approximately 100 cells were visualized with a similar frequency of double-positive cells as observed in (d). No single-positive cells were detected. (f) Analysis of DNAs as in (c) using murine lymph node lymphatic endothelial cells. (g) BMDCs were incubated with ova-psDNA (conjugated), ova plus psDNA (unconjugated), or PBS for 1, 3, and 7 days prior to adding OT-1 T cells labeled with violet proliferation dye. T cells and BMDCs were co-cultured at a ratio of 1:10 for 3 days. (h) Quantification of (g) using the percent divided calculation described in the Materials and methods. Experiments were performed three times with 3–5 wells per sample with similar results. Error bars represent SEM. Asterisks denote sample significant amounts relative to the negative control (p<0.05 Wilcoxon rank-sum test). Exact p-values are as follows: day 1 psDNA:ova-psDNA p=0.008, psDNA:untreated p=0.016, ova-psDNA:untreated p=0.016; day 3 psDNA:ova-psDNA p=0.008, psDNA:untreated p=0.016, ova-psDNA:untreated p=0.016; day 7 psDNA:ova-psDNA p=1, psDNA:untreated p=0.400, ova-psDNA:untreated p=0.400. n.d.: none detected.

The online version of this article includes the following source data and figure supplement(s) for figure 1:

**Source data 1.** Antigen tags and other oligonucleotide sequences used in qPCR and single-cell experiments.
**Figure supplement 1.** Visualization of antigen and DNA in different cell types.
**Figure supplement 2.** DNA barcode is not retained in cell media over time.

supplement 1d–f). Using a more phagocytic cell, bone marrow-derived macrophages, we observed nearly all macrophages phagocytosed the ova-psDNA at day 1 and found the ova and psDNA within the same cell (*Figure 1—figure supplement 1g*). In macrophages given ova-psDNA 7 days prior, we detected only ova protein (*Figure 1—figure supplement 1g*), potentially resulting from high levels of endonucleases found within the lysosome of macrophages (*Krieser et al., 2002*; *Nagata, 2007*).

To determine whether conjugation of psDNA to ova affected ova processing and presentation, we measured BMDC presentation of ova-derived SIINFEKL peptide by co-culture with SIINFEKL-specific OT1 T cells. BMDCs given ova-psDNA induced significantly more proliferation of OT1 T cells than unconjugated ova (*Figure 1g, h*), suggesting enhanced activation of BMDCs upon encounter with ova-psDNA conjugates. Furthermore, we detected pDNA and psDNA in BMDC culture media at 1 day after addition but not at later time points, confirming that ova-psDNA conjugates are processed and not released by BMDCs after phagocytosis (*Figure 1—figure supplement 2a, b*). Finally, ova-psDNA conjugates led to increased OT1 proliferation relative to ova plus psDNA (unconjugated), showing that ova-psDNA conjugates are immunostimulatory (*Figure 1g, h*) and consistent with studies showing conjugation of antigens to RNA or DNA induce TLR7 (RNA) or TLR9 (DNA) signals that lead to prolonged antigen presentation (*Xu and Moyle, 2018*). Addition of polyI:C and anti-CD40 to BMDCs with ova also elicited robust OT1 proliferation, demonstrating that TLR activation on the BMDCs is required for efficient cross-presentation to T cells (*Figure 1—figure supplement 2c*).

We next asked whether vaccination with ova-psDNA conjugates elicits a T cell response in vivo. We compared antigen-specific T cell responses in mice vaccinated with a mixture of ova-psDNA and polyI:C/αCD40 to its individual components (ova, psDNA, polyI:C, and polyI:C/αCD40; *Figure 2a*, *Figure 2—figure supplement 2a, b*) and—consistent with the differences in OT1 proliferation we saw in vitro—found that T cell responses to ova-psDNA were greater than either ova with polyI:C, ova with polyI:C/αCD40, or a mixture of unconjugated ova and psDNA (*Figure 2b*). Interestingly, ova-psDNA conjugate combined with polyI:C/αCD40 did not significantly enhance the T cell response beyond ova-psDNA alone (*Figure 2b*). T cells stimulated by ova-psDNA produced significantly more IFNγ than any other vaccination strategy even in the absence of ex vivo SIINFEKL peptide stimulation, indicating prolonged and active presentation of ova-psDNA (*Figure 2c, d*). Together, these data show that ova-psDNA conjugates elicit antigen-specific T cell responses independent of polyI:C/αCD40. These findings are consistent with TLR9-dependent immune responses

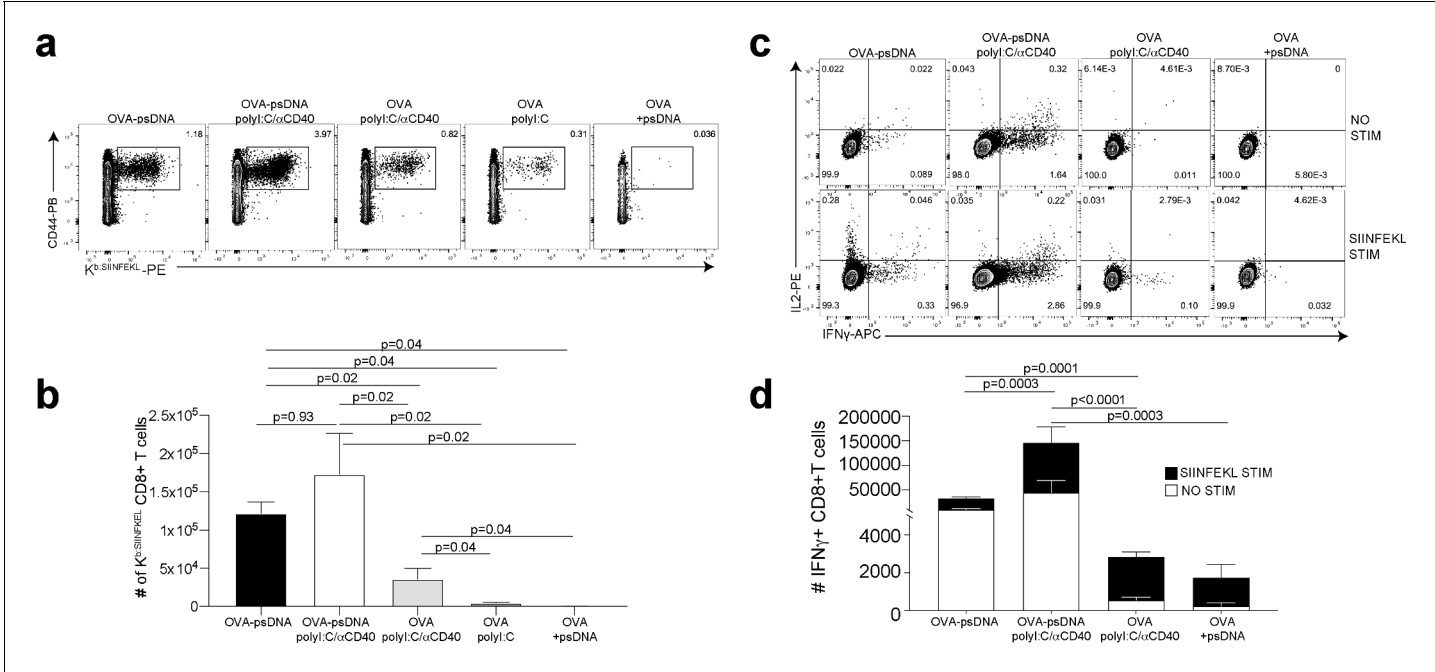

**Figure 2.** Antigen-psDNA conjugates elicit a robust immune response in vivo. (**a**) Mice were immunized in the footpad with ovalbumin (ova) alone or ova-psDNA with or without polyI:C/aCD40 or polyI:C. After 7 days, draining popliteal lymph nodes (LNs) were harvested and cells were stained and gated as B220-, CD3+, CD8+, CD44+, and OVA257 Kb SIINFEKL-specific tetramer to measure antigen-specific CD8 T cell responses. (**b**) Quantification of SIINFEKL-specific CD8 T cells within the LN (data from **a**). Experiment was performed three times; shown is combined data from at least three mice per group, per experiment. p-Values were calculated using a two-stage step-up method of Benjamini, Krieger, and Yekutieli and did not assume consistent standard deviation. Error bars represent standard error of the mean (SEM). (**c**) As in (**a**) and (**b**) except cells were restimulated with SIINFEKL peptide for 6 hr ex vivo in the presence of brefeldin A, then stained for IFNγ and IL-2. (**d**) Quantitation of IFNγ-positive CD8+ T cells with or without peptide stimulation in the draining LN. Experiment was performed three times; shown is combined data from at least three mice per group, per experiment. p-Values were calculated using a two-stage step-up method of Benjamini, Krieger, and Yekutieli and did not assume consistent standard deviation. Error bars represent SEM.

The online version of this article includes the following figure supplement(s) for figure 2:

**Figure supplement 1.** Gating strategies.

**Figure supplement 2.** Vaccinia plus ovalbumin (ova)-DNA conjugate induces ova-specific T cell response and archiving.

elicited by psDNA (*Baek et al., 2001*; *Coffman et al., 2010*; *Vollmer et al., 2004*), similar to DC presentation of conjugates of ova demonstrated with other TLR agonists (*van Montfoort et al., 2009*) and other subcutaneously administered ova-TLR conjugate vaccine platforms (*Xu and Moyle, 2018*).

We previously showed that a vaccination strategy comprising soluble antigen and vaccinia virus (VV; Western Reserve) induced robust antigen archiving that lasts longer than those using polyI:C/α CD40 adjuvant (*Kedl et al., 2017*). To evaluate antigen-psDNA performance during an active infection, we determined T cell responses after vaccination by comparing individual components with mixtures of ova, VV, ova-pDNA, or ova-psDNA. Subcutaneously administered ova-psDNA alone again elicited a T cell response (*Figure 2*, *Figure 2—figure supplement 2a*), and addition of VV to ova-psDNA conjugate moderately increased T cell responses compared to ova-psDNA alone, similar to what we observed with ova-psDNA/polyI:C/αCD40 (*Figure 2*, *Figure 2—figure supplement 2b*). Finally, we examined the cell-type specificity of ova-psDNA dissemination in vivo. Mice were vaccinated with mixtures of (i) ova-psDNA and VV or (ii) ova-psDNA and polyI:C/αCD40, and levels of ova-psDNA were quantified by PCR in both leukocytes and SCs (fractionated by CD45 expression) in the draining LNs. We found that CD45- SCs had high amounts of ova-psDNA, but not ova-pDNA, corresponding to increased inflammation (*Tamburini et al., 2014*), whereas CD45+ leukocytes had very low levels of ova-psDNA or ova-pDNA 7 days after vaccination (*Figure 2—figure supplement 2c*). These data recapitulate our previous demonstration of durable antigen retention by CD45- SCs

(*Kedl et al., 2017*; *Tamburini et al., 2014*), confirming that ova-psDNA, but not ova-pDNA, is a faithful tracking device for antigen archiving in vivo.

## Molecular tracking of antigen during the immune response to vaccination

Given the ability of the antigen-psDNA conjugates to induce a robust immune response in vivo (*Figure 2*) and our ability to use the psDNA as a measure of protein antigen levels (*Figure 1*), we used the antigen-psDNA conjugate as a 'molecular tracking device' to understand the distribution of the protein antigen in the LN following this vaccination. To determine whether we could identify if cells acquire and archive (*Tamburini et al., 2014*) antigens following antigen-psDNA, we vaccinated mice subcutaneously with an equimolar mixture of uniquely barcoded ova-psDNA conjugate, unconjugated psDNA, and unconjugated pDNA (unprotected phosphodiester backbone) with VV (as in *Figure 2—figure supplement 2c*), and evaluated antigen distribution (via psDNA abundance) in the LN at early (2 days) and late (14 days) time points. At each time point, single-cell suspensions were prepared from draining popliteal LNs and divided into SC (by depleting CD45+ cells) or lymphocyte populations (by flow sorting for CD11c, CD11b, and B220 markers; *Figure 2—figure supplement 1b*). To enrich for myeloid cell populations but maintain representation of other cell types, CD11c+, CD11b+, B220+, and ungated live cells were mixed at a 4:4:1:1 ratio, respectively. These cell populations were analyzed by single-cell mRNA sequencing, measuring both mRNA expression and the quantity of psDNA in each cell using unique molecular identifiers (*Islam et al., 2014*; *Figure 3*).

We recovered a total of 800 cells in the CD45- fraction and 8187 cells in the CD45+ fraction at the 2-day time point. We recovered more CD45- cells (6372 CD45-; 4840 CD45+) at the 14-day time point likely due to expansion and proliferation of the LN stroma (*Tamburini et al., 2014*; *Lucas et al., 2018*; *Lucas and Tamburini, 2019*). We classified cell types using an automated approach (*Fu et al., 2020*), comparing measured mRNA expression patterns to reference data sets for DCs (*Brown et al., 2019*; *Miller et al., 2012*), fibroblastic reticular cells (FRC)s (*Rodda et al., 2018*), and LECs (*Fujimoto et al., 2020*; *Kalucka et al., 2020*; *Xiang et al., 2020*; *Figure 3—source data 1*). As expected, the CD45+ fraction contained DCs, monocytes, T cells, and B cells (*Figure 3a, b, d, e*), while the CD45- fraction contained SCs, including LECs, BECs, epithelial cells, and fibroblasts (*Figure 3g , h, j, k*). We did not recover VV mRNAs in cells at either time point, possibly due to viral clearance or a failure to recover infected, apoptotic cells in the live/dead selection (*Figure 2—figure supplement 1b*).

We first examined the dynamic changes of myeloid populations in the LN. We detected conventional DCs, including cDC1 and cDC2 (*Figure 3a–c*), which develop from a common DC precursor upon expression of FMS-like tyrosine kinase 3 ligand (Flt3L) (*Guilliams et al., 2014*). LN-resident and migratory cDCs can be distinguished by expression of cell-type-specific transcription factors including basic leucine zipper transcription factor (Batf3) and interferon regulatory factor (IRF8) (cDC1) (*Aliberti et al., 2003*; *Hildner et al., 2008*; *Tsujimura et al., 2003*) or IRF4 and Notch (cDC2) (*Lewis et al., 2011*; *Schlitzer et al., 2013*). These cDC types are also typically classified based on expression of CD11c, Zbtb46, and chemokine XC receptor 1 (cDC1 are XCR1+, cDC2 are XCR1-) (*Guilliams et al., 2014*; *Bachem et al., 2012*). cDC2s are further categorized as either Tbet-dependent and anti-inflammatory (cDC2A) or RORγt-dependent and pro-inflammatory (cDC2B) (*Brown et al., 2019*).

As expected, at day 2 we identified a large population of LN-resident cDC2B (cDC2 Tbet-) cells harboring ova-psDNA (*Brown et al., 2019*). However, we did not find any cDC2A (cDC2 Tbet+) cells, consistent with their role in anti-inflammatory processes (*Brown et al., 2019*). The myeloid populations contained CCR7[hi] cDCs (n = 3432; 42% of total), which we classified as migratory DCs. This migratory DC population included Langerhans cells (n = 285; 3.5% of total), migratory cDC1s (n = 593; 7.2% of total), and migratory cDC2s (n = 2554; 31% of total) (*Miller et al., 2012*), migrating from the dermis (*Figure 3b*). At day 14, we identified a population of LN-resident cDC2 Tbet+ cells (*Figure 3e*) consistent with resolution of the immune response (*Brown et al., 2019*). As cDC2 Tbet+ cells are thought to be anti-inflammatory, these data suggest that the immune response is being quelled (*Figure 3e*). We also found a group of Siglec-H+ DCs, a cDC progenitor population (*Brown et al., 2019*; *Figure 3d, e*).

Using unique barcodes, we quantified the amount of ova-psDNA, psDNA, and pDNA across cell types. Levels of ova-psDNA molecules spanned four orders of magnitude, ranging up to $10^4$ unique

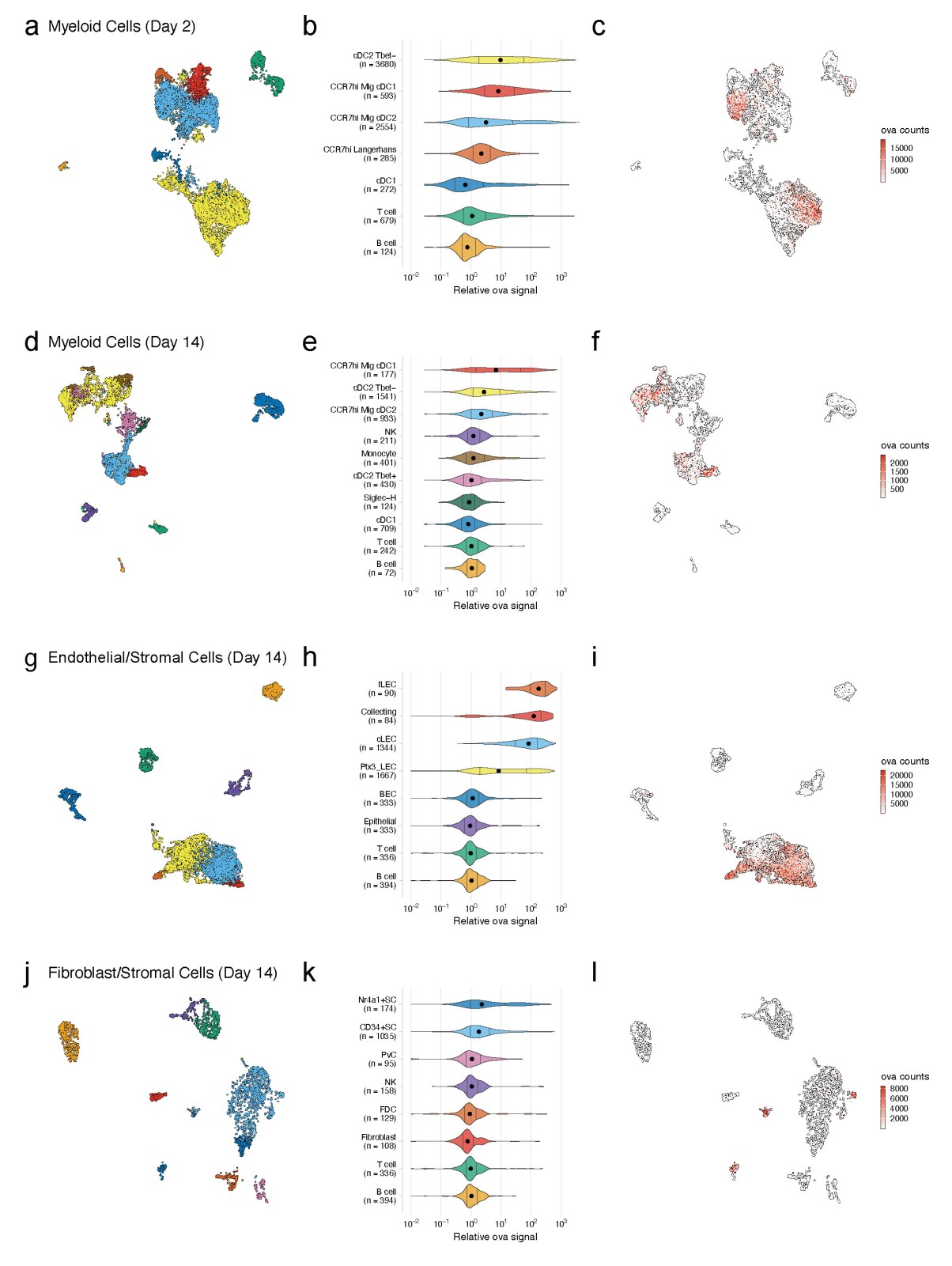

**Figure 3.** Dynamic acquisition of antigen-psDNA conjugates in lymph node tissue. (a, d, g, j) Uniform manifold approximation and projections (UMAPs) are shown for dendritic cells (DCs) (a, d), lymphatic endothelial cells (LECs) (g), and fibroblastic reticular cells (FRC)s (j) at day 2 (a) and day 14 (d, g, j). (b, e, h, k) Relative ovalbumin (ova) signal was calculated by dividing antigen counts for each cell by the median antigen counts for T and B cells. Signals are plotted on $log_{10}$ scale; black dots indicate median values, and vertical lines denote quartiles. Statistical comparisons between each pair of

*Figure 3 continued on next page*

*Figure 3 continued*

groups are available in **Figure 3—source data 1**. (c, f, i, l) unique moleular identifier (UMI)-adjusted antigen counts are displayed on UMAPs for each cell type.

The online version of this article includes the following source data and figure supplement(s) for figure 3:

**Source data 1.** Comparison of relative ovalbumin (ova) signal for cell types shown in **Figure 3—figure supplements 3**, **4b,** and **5b**.
**Figure supplement 1.** Detection of DNA barcode requires conjugation to ovalbumin (ova).
**Figure supplement 2.** Antigen counts were independent of total mRNA counts.
**Figure supplement 3.** Lymphatic endothelial cell (LEC) types associated with high antigen counts 2 days after vaccination.
**Figure supplement 4.** Antigen is held by PD-L1/ICAM1 high lymphatic endothelial cells (LECs).
**Figure supplement 5.** FRC cell types with high antigen counts at day 2 post vaccination.

molecules and depending on the cell types and time point (*Figure 3c, f, i, l*). In contrast to the large range of ova-psDNA across cell types, unconjugated psDNA and pDNA were largely undetectable, indicating that antigen conjugation is required for cell acquisition (*Figure 3—figure supplement 1*). Consistent with our previous studies (*Kedl et al., 2017*), we did not detect antigen-psDNA at appreciable levels in T cells or B cells (*Figure 3*), and because these cell types were captured in both our CD45- and CD45+ samples, we used their median antigen levels to normalize antigen counts in other cell types across captures. We considered the trivial case wherein variation in antigen levels is explained by total mRNA abundance; these variables are uncorrelated in SC types and weakly correlated in cDC subtypes, possibly reflecting activation status (*Figure 3—figure supplement 2*).

At the early day 2 time point, LN-resident cDC2s contained high levels of antigen-psDNA, consistent with studies of antigens administered with alum (*Gerner et al., 2017*; *Figure 3b, c*). In addition, we found significantly higher levels of antigen in cDC2 Tbet-, migratory CCR7$^{hi}$ cDC2s, and migratory CCR7$^{hi}$ cDC1s (*Figure 3b, c*, *Figure 3—source data 1*), with an average of approximately sevenfold more antigen than T/B cells. At the later time point, migratory cDC1 cells contained the most antigen, consistent with previous studies (*Kedl et al., 2017*; *Figure 3e*). In addition, Tbet- and CCR7$^{hi}$ migratory cDC2s contained moderate levels of antigen, up to threefold more than T/B cells, but had lower amounts of antigen relative to day 2 (*Figure 3b, c, e, f*, *Figure 3—source data 1*). At the late time point, we did not detect significant amounts of antigen in LN-resident cDC1s, Tbet+ cDC2s, Siglec-H+ cells, or monocytes (*Figure 3e*).

We next examined antigen levels in the LN SC populations (*Figure 3g–l*, *Figure 3—source data 1*). Endothelial cells in the LN are classified by their association with blood or lymphatic vasculature; both are required for circulation and trafficking of immune cells to the LN. The blood vasculature circulates naive lymphocytes to the LN, and the lymphatic vasculature transports immune cells from the peripheral tissue including dermal DCs and memory T cells. We used an automated approach (*Fu et al., 2020*) that uses correlation between reference and measured gene expression profiles to assign unknown cell types to subtypes defined by previous studies. While strong correlation reflects a good match between reference and query profiles, high correlation between multiple reference LEC subtypes (*Fujimoto et al., 2020*; *Kalucka et al., 2020*; *Xiang et al., 2020*) and changes in expression induced by antigen acquisition made definitive cell-type assignments challenging (*Figure 3—figure supplement 3a–c*). Notwithstanding these issues, we classified LEC subsets based on the highest correlation values to reference cell types (*Figure 3—figure supplement 3d, e*; *Xiang et al., 2020*) and identified three LEC subtypes (*Fujimoto et al., 2020*; *Kalucka et al., 2020*; *Xiang et al., 2020*) including Ptx3 LECs, ceiling LECs, and Marco LECs with high levels of antigen at the early time point (*Figure 3—figure supplement 3*). At the late time point, expansion and proliferation of LN SCs contributed to larger populations of cells including floor LECs, collecting LECs, ceiling LECs, Ptx3 LECs (*Kalucka et al., 2020*), and BECs (*Figure 3h*; *Malhotra et al., 2012*).

At the day 14 time point, several LEC subtypes maintained high antigen levels (*Figure 3h*, *Figure 3—source data 1*). Floor LECs had uniformly high amounts of antigen. Median levels of ova-psDNA were detected in collecting, Ptx3, and ceiling LEC populations that were significantly higher than B/T cells; however, cells in these groups contained a range of antigen with both high and low populations. We hypothesized that this variability stems from the physical location of the LECs within the LN and their access to trafficking antigen. Using a fluorescently labeled ova with polyI:C/α CD40[12], we confirmed that fluorescent antigen amounts are highest on subcapsular LECs as identified by surface expression of PD-L1 and ICAM1 2 weeks after immunization, similar to ova-psDNA

vaccination (*Lucas et al., 2018*; *Cohen et al., 2014*; *Figure 3—figure supplement 4*). Together, our findings suggest that antigen first passes through the sinus followed by the cortex and medulla. These data also suggest that populations of LECs with less antigen could be a result of how the antigen travels through the LN or mechanisms of antigen release over time.

Similar to the endothelial cell population, the number and types of non-endothelial SCs increased at the later time point after immunization. Non-endothelial SCs in the LN are classified by their location in the LN into T-zone reticular cells (TRC), marginal reticular cells (MRCs), follicular dendritic cells (FDCs), and perivascular cells (PvCs) (*Rodda et al., 2018*). Recently, additional subsets were identified including Ccl19$^{lo}$ TRCs located at the T-zone perimeter, Cxcl9$^+$ TRCs found in both the T-zone and interfollicular region, CD34$^+$ SCs found in the capsule and medullary vessel adventitia, indolethylamine N-methyltransferase$^+$ SCs found in the medullary chords, and Nr4a1+ SCs (*Rodda et al., 2018*).

At the early time point, the Cxcl9+ TRCs and CD34+ SCs (*Rodda et al., 2018*) had high amounts of antigen (~10-fold relative to T/B cells) (*Figure 3—figure supplement 5*). At the late time point, we detected CD34+ SCs, Nr4a1+ SCs, FDCs, and PvCs (*Figure 3k*). Only the CD34+ and Nr4a1+ SCs contained significant amounts of antigen (*Figure 3k*, *Figure 3—source data 1*). Interestingly, the CD34+ SCs are adjacent to ceiling LECs and the Nr4a1+ SCs are found in the medullary chord and medullary sinus, which are lined by medullary LECs. These findings may suggest potential antigen exchange mechanisms between LECs and SCs that have yet to be defined. We found little antigen in PvCs or FDCs (*Figure 3k*, *Figure 3—source data 1*).

Finally, these data provided insight into antigen transfer between SCs and DCs, a process important for enhanced protective immunity (*Kedl et al., 2017*; *Tamburini et al., 2014*). We previously showed that archived antigen obtained from the polyI:C/anti-CD40-based vaccine is transferred from LECs to migratory *Batf3*-dependent cDC1s 2 weeks after infection (*Kedl et al., 2017*). Here, we confirm that with the ova-psDNA vaccine CCR7$^{hi}$ migratory cDC1s had the highest amount of antigen 2 weeks after vaccination (*Figure 3e*, *Figure 3—source data 1*; *Kedl et al., 2017*). Together, these data validate the use of molecular tracking devices by corroborating previous studies of antigen trafficking with other vaccination strategies and identify new cells types that dynamically acquire antigen during infection.

## Gene expression signatures associated with antigen acquisition by DCs

We next leveraged the variation in antigen levels across cell types (*Figure 3b, e, h, k*) to identify gene expression signatures associated with high levels of antigen that would validate our approach. We classified cells as 'antigen-high' and 'antigen-low' using a two-component mixture model and identified marker genes associated with each class (*Figure 4a, b*). To validate this approach, we evaluated the DC populations as genes associated with phagocytosis and activation have been established (*Miller et al., 2012*; *Breuilh et al., 2007*; *Bune et al., 2001*; *Figueiredo et al., 2018*; *Gschwandtner et al., 2019*; *Hirano et al., 2007*; *Jin et al., 2020*; *Lämmermann and Kastenmüller, 2019*; *Mancardi et al., 2008*; *PrabhuDas et al., 2017*; *Sinclair, 1999*). DC populations generally contained lower antigen levels that were variable across subtype (*Figure 3*). We classified antigen-low and antigen-high cells for each subtype. Among the subtypes with significant amounts of antigen, Tbet- cDC2 cells had the highest antigen levels and largest differences in gene expression (277 genes in antigen-high cells, *Figure 4*, *Figure 4—source data 1*), consistent with cDC2s acting as the primary cell type of antigen uptake following protein (*Gerner et al., 2017*).

At the early time point, genes upregulated in antigen-high DCs confirmed DC activation (*Figure 4—source data 1*). Antigen-high cDC2 Tbet- cells upregulated genes *Ccl2* and *Cxcl2* (consistent with active recruitment of inflammatory cells; *Gschwandtner et al., 2019*; *Lämmermann and Kastenmüller, 2019*), *Msr1* (consistent with antigen scavenging; *PrabhuDas et al., 2017*), as well as *Pkm*, *Lgals3*, and *Mif* (consistent with DC-T cell responses and DC differentiation during inflammation; *Breuilh et al., 2007*; *Figueiredo et al., 2018*; *Jin et al., 2020*; *Figure 4c, d*).

At the late day 14 time point, the highest antigen counts were found in the migratory cDC1 population, consistent with a role for migratory cDC1s in archived antigen acquisition from LECs (*Kedl et al., 2017*; *Figure 3e, f*). Among the genes highly expressed by the antigen-high CCR7$^{hi}$ migratory cDC1 population were *Ccl5* and *Fscn1* (*Figure 4—source data 1*). Consistent with these DCs being involved in archived antigen presentation, *Ccl5* (also known as RANTES) regulates CD8 T cell responses during chronic viral infection (*Crawford et al., 2011*) and *Fscn1*, an actin binding

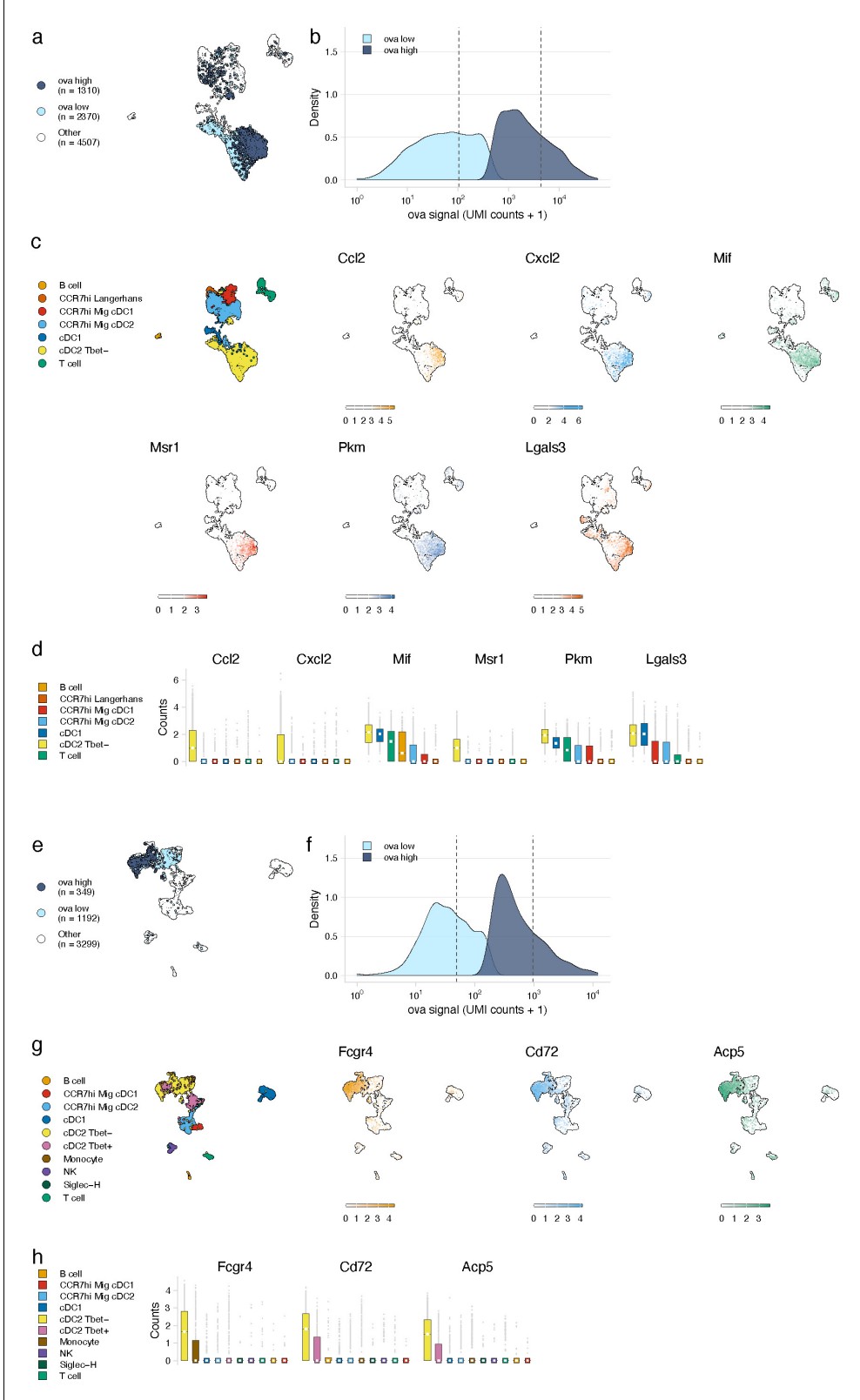

**Figure 4.** Antigen-based classification of dendritic cells (DCs) and validation of genes associated with DC activation. (a, e) Day 2 (a) and day 14 (e) cDC2 Tbet- cells containing low and high antigen counts were identified using a two-component mixture model. A uniform manifold approximation and projection (UMAP) is shown for ovalbumin (ova)-low and ova-high cells. Cell types not included in the comparison are shown in white (other). (b, f) The distribution of ova antigen counts is shown for ova-low and ova-high cDC2 Tbet- cells. Dotted lines indicate the mean counts for each population.

*Figure 4 continued on next page*

*Figure 4 continued*

Identification of genes associated with ova-low and ova-high for each cell type is available in *Figure 4—source data 1*. (c, g) UMAPs show the expression (log-normalized counts) of top markers associated with ova-high cDC2 Tbet- cells. (d, h) Expression (log-normalized counts) of antigen-high markers in each cell type.

The online version of this article includes the following source data for figure 4:

**Source data 1.** Genes associated with ovalbumin (ova)-high cells for dendritic cell (DC), FRC, and lymphatic endothelial cell (LEC) subtypes.

protein, regulates cell migration of mature DCs via podosome formation (*Yamakita et al., 2011*). Similar to the day 2 time point, among subtypes with significant amounts of antigen, Tbet- cDC2 populations showed the greatest differences in gene expression between antigen-high and -low cells (230 genes in antigen-high cells; *Figure 4e, f*, *Figure 4—source data 1*). Genes upregulated in antigen-high Tbet- cDC2s included *Fcgr4*, which is involved in phagocytosis, antigen presentation, and proinflammatory cytokine production (*Hirano et al., 2007*; *Mancardi et al., 2008*), and *CD72* and *Acp5*, which are important for the inflammatory response and pathogen clearance (*Bune et al., 2001*; *Sinclair, 1999*; *Figure 4g, h*). Collectively, these genes evoke specific processes in DC subsets required for the immune response; it remains to be determined whether they are specifically associated with LEC-DC antigen exchange or storage of antigens within DCs.

## Gene expression signatures associated with antigen archival by LECs

We next evaluated the LEC population to determine whether our classification approach could identify genes involved in antigen archiving. We applied the classifier to LECs as a population and found large numbers of antigen-high-floor, collecting, and ceiling LECs (*Figure 5c*). Ptx3 LECs comprised a mixture of antigen-low and antigen-high cells, but there was a larger fraction of Ptx3 LECs with low antigen (*Figure 5c*). There were less antigen-low LECs compared to antigen-high LECs overall (34% of total), suggesting that antigen archiving may be specific to LECs in general rather than attributable to a specific LEC subset (*Figure 5*).

Using this classification approach, we identified 142 mRNAs that were significantly changed in antigen-high or antigen-low LECs (*Figure 5—source data 1*). *Prox1,* while expressed by all LECs identified, was highly expressed in antigen-high LECs, independent of the LEC type (*Figure 5d, e*). *Prox1* is a transcription factor required for LEC differentiation from BECs and defines LEC identity via regulation of *Vegfr3*, *Pdpn,* and *Lyve-1* (*Harvey et al., 2005*; *Hong et al., 2002*; *Wigle and Oliver, 1999*). *Prox1* upregulation in antigen-high LECs indicates it may also transcriptionally regulate processes involved in antigen archiving.

Upregulation of *Cavin1* and *Cavin2* by antigen-high LECs suggested that caveolar endocytosis may contribute to antigen acquisition by LECs, consistent with LEC dynamin-mediated transcytosis in vitro (*Triacca et al., 2017*; *Figure 5d, e*). *Cavin2* appears more specific to LECs than *Cavin1*, which is also upregulated by BECs, suggesting that *Cavin2* mediates endocytosis specifically in endothelial cells of the lymphatic lineage. Based on *Cavin2* gene expression, it appears that this process may be most active in ceiling LECs (*Figure 5e*). To confirm this finding, we asked whether inhibition of the caveolin pathway with nystatin impaired endocytosis of fluorescent antigen in mice vaccinated with polyI:C/αCD40. We found a significant decrease in antigen acquired by LECs in the nystatin treatment group 24 hr after administration of fluorescent antigen with this vaccine regimen (*Figure 5f*), affirming the utility of molecular tracking devices for identifying genes involved in the process of antigen acquisition or archival that are not necessarily specific to antigen-psDNA conjugates.

Finally, expression of Stabilin-1 (*Stab1)* and Stabilin-2 (*Stab2)* is increased in antigen-high LN endothelial cells, suggesting that scavenging pathways are required for the acquisition of antigen-psDNA conjugates after vaccination. *Stab2* is uniquely expressed by LECs in the LN and not by BECs (*Malhotra et al., 2012*), and *Stab1* and *Stab2* act as receptors for internalization of antisense oligonucleotides with phosphorothioate linkages in liver endothelial cells and Kupffer cells (*Miller et al., 2016*). However, we did not find significant amounts of unconjugated psDNA in LECs (*Figure 3—figure supplement 1*), indicating that *Stab1/Stab2* are upregulated as part of an antigen scavenging or trafficking program initiated in LECs upon antigen acquisition during infection.

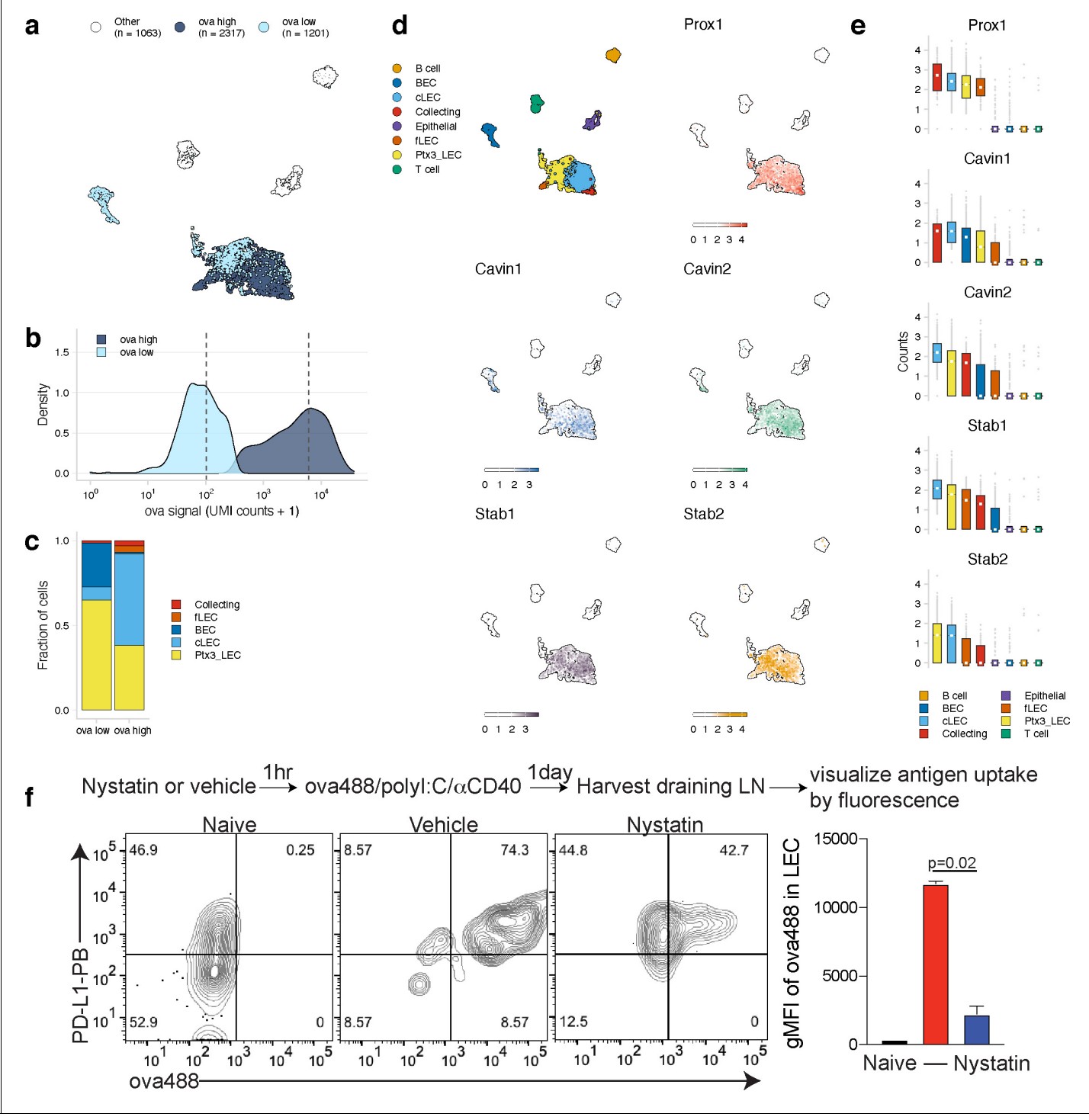

**Figure 5.** Antigen-based classification of lymphatic endothelial cells (LECs) and identification of marker genes. (a) Day 14 LECs were classified into antigen-high and antigen-low using a two-component Gaussian mixture model. A uniform manifold approximation and projection (UMAP) is shown for antigen-low and antigen-high cells. T cells, B cells, and epithelial cells are shown in white (Other). (b) Distribution of antigen counts for antigen-low (light blue) and antigen-high (dark blue) cells. Dotted lines indicate mean counts for each population. Identification of genes associated with ovalbumin (ova)-low and ova-high for each cell type is available in *Figure 5—source data 1*. (c) The fraction of cells belonging to each LEC type for antigen-low and antigen-high populations. (d) UMAPs show expression of genes significantly enriched in the antigen-high population (scale is log-normalized counts). (e) Expression (log-normalized counts) of antigen-high markers in each cell type. (f) Mice were injected in the footpad with nystatin (dose) and 1 hr later ova488/polyI:C/αCD40. After 24 hr, mice were euthanized and draining popliteal lymph node (LN) removed, stained for LEC markers (CD45, PDPN, CD31, PDL1), and gated as in *Figure 3—figure supplement 4*. Shown are representative flow plots and quantification of

*Figure 5 continued on next page*

*Figure 5 continued*

geometric mean fluorescence intensity (gMFI) from naive (black bar), vehicle control (red bar), and nystatin treated (blue bar). Three mice per group were evaluated, and experiment was performed three independent times with similar results. Nystatin treatment reduces ova488 signal in LECs relative to vehicle (p=0.02; Wilcoxon rank-sum test). Error bars indicate standard error of the mean (SEM).

The online version of this article includes the following source data for figure 5:

**Source data 1.** Genes associated with ovalbumin (ova)-low and ova-high cells for dendritic cells (DCs), FRCs, and lymphatic endothelial cells (LECs).

## Discussion

Our development of a 'molecular tracking device' enabled tracking of antigen throughout the LN to specific cell types that acquire and archive antigens following subcutaneous immunization. Previous studies used canonical surface markers to track antigen by microscopy and flow cytometry; instead, our approach simultaneously defines cell type by gene expression and quantifies the acquired antigen. The molecular tracking device includes phosphorothioate DNA conjugation, which provides a combined TLR-antigen delivery system to study antigen distribution at time points beyond the lifetime of antigen-fluorophore conjugates and provided a map of cell types involved in antigen-psDNA acquisition and retention.

Our approach expands upon our previous studies with other vaccine regimens that induce antigen archiving and cell types that enhance protective immunity. Both here and in our previous studies, we found that whereas LECs archive antigen, migratory DCs passing through the lymphatic vasculature are required to retrieve and present archived antigen to memory CD8 T cells derived from the initial infection or immunization (*Eisenbarth, 2019*). Using an antigen/polyI:C/αCD40 vaccine regimen, we determined that antigen exchange from LECs to DCs and subsequent DC presentation yields memory CD8 T cells with robust effector function during infectious challenge. The studies included here predict the same outcome as both LECs and migratory DCs were detected with ova-psDNA at the late time point. Several recent reports defined LEC and non-endothelial SC subsets within the LN (*Rodda et al., 2018*; *Fujimoto et al., 2020*; *Kalucka et al., 2020*; *Xiang et al., 2020*). By combining our molecular tracking device with these reference cell types, we found that non-endothelial SC types acquire foreign antigens including CD34+ SCs, which neighbor subcapsular sinus LECs in the tissue (*Rodda et al., 2018*). These findings suggest that the interstitial pressure created by subcutaneous vaccination allows antigens to pass through the tissue directly to the LN capsule, bypassing the lymphatic capillaries. Intriguingly, bypass of lymphatic capillaries may still lead to LEC acquisition of antigens from the CD34+ SCs via SC-LEC exchange. Such a mechanism would encourage future LEC-DC interactions and provide a benefit to protective immunity.

Molecular tracking devices provide a measure of cell state orthogonal to gene expression, which we leveraged to identify candidate pathways involved in antigen-psDNA acquisition (*Figure 4*). We show that the caveolin pathway is upregulated in antigen-high LECs and demonstrate this pathway is involved in antigen acquisition in vivo following vaccination with fluorescent ova/polyI:C/αCD40 via pharmacological inhibition of caveolar endocytosis (*Figure 4f*). Genes uniquely expressed by LECs such as *Prox1*, *Cavin2*, and *Stab2* (*Miller et al., 2012*; *Heng et al., 2008*; *Malhotra et al., 2012*) represent targets for further manipulation of antigen archiving by LECs.

The psDNA component of the tracking device elicits an immune response similar to other TLR-antigen conjugate vaccines (*Oh and Kedl, 2010*; *Oh et al., 2012*), likely due to antigen-psDNA stability within DCs that causes prolonged antigen presentation in the cells that acquire the antigen (*van Montfoort et al., 2009*; *Xu and Moyle, 2018*). This effect is illustrated by increased IFNγ production in the absence of ex vivo peptide stimulus (ova-psDNA compared to unconjugated ova; *Figure 2*). Prolonged antigen presentation better replicates an infection wherein levels of viral or bacterial antigen rise over the duration of infection. However, in other applications it may be helpful to limit the immunoreactivity of the antigen-psDNA via cytosine methylation (*Hemmi et al., 2000*) or backbone modification (*Lange et al., 2019*). While many of the experiments we performed with the ova-psDNA were consistent with our experiments using antigen-TLR conjugates or TLR/CD40-based vaccines, it is likely that this type of vaccine interacts with different cell types and utilizes different mechanisms for antigen acquisition and retention. These mechanisms are currently under active investigation and may be more generalizable in the absence of TLR9.

A caveat of our studies concerns the dynamic stability of molecular tracking devices in tissue. Multiple detection methods showed that the protein and DNA components of our conjugates co-localize in LECs (*Figure 1f*, *Figure 1—figure supplement 1c*) and bone marrow-derived DCs (*Figure 1c, d, g, h*), and unconjugated psDNA was untrackable both in vitro or in vivo (*Figure 1c, f*, *Figure 3— figure supplement 1*), indicating that psDNA is not readily taken up by cells. However, flow cytometry analysis of conjugates in BMDM indicates that DNA degradation may precede protein degradation (*Figure 1—figure supplement 1g*). With that said, it remains possible that acquisition of molecular tracking devices by certain cell types leads to decoupling of the individual components after which they could be independently transferred to other cells via trogocytosis or other mechanisms of membrane transfer (*Alegre et al., 2010*). Closer evaluation of the protein-DNA complex in vivo over time will be important to determine how accurately detection of the DNA via single-cell sequencing reflects the movement of the protein-DNA complex. Future experiments will address the dynamics of conjugate stability across multiple cell types to quantify the low levels of unconjugated components and better define the limitations of molecular tracking devices in studying protein degradation intermediates.

Molecular tracking devices will enable new approaches to study molecular dissemination in vivo. To date, protein-DNA conjugates have been deployed in single-cell mRNA sequencing experiments for ex vivo staining applications (e.g., CITE-seq; *Islam et al., 2014*). Our study lays the groundwork for molecular tracking devices involving protein, antibody, drug, or pathogens conjugated to nuclease-resistant, barcoded oligonucleotides that are stable during transit through mouse tissues. The approach naturally extends to understanding how multiple different antigens might be processed (using unique DNA barcodes) and enables new studies to manipulate antigen archiving to improve vaccines, vaccine formulations, and prime-boost strategies. Moreover, the oligonucleotide portion of the tracking device should enable analysis of its distribution in cells by in situ hybridization or intact tissue by spatial transcriptomics (*Eng et al., 2019*; *Rodriques et al., 2019*; *Ståhl et al., 2016*), obviating the need for antibody-mediated detection of antigen.

## Materials and methods

### Key resources table

| Reagent type (species) or resource | Designation | Source or reference | Identifiers | Additional information |
|---|---|---|---|---|
| Antibody | Anti-mouse CD40 (Rat monoclonal) | BioXcell | Cat#BE0016-2 | |
| Antibody | Anti-ovalbumin (rabbit monoclonal) | Abcam | Ab181688 | 1:100 |
| Antibody | Anti-rabbit IgG PE (Donkey polyclonal) | Biolegend | Cat# 406421 RRID:AB_2563484 | 1:100 |
| Antibody | Anti-mouse CD45 BV510 (Rat monoclonal) | Biolegend | Cat#103138 RRID:AB_2563061 | 1:300 |
| Antibody | Anti-mouse CD45 PE (Rat monoclonal) | Biolegend | Cat#103106 RRID:AB_312971 | 1:300 |
| Antibody | Anti-mouse podoplanin APC (Hamster monoclonal) | Biolegend | Cat#127410 RRID:AB_10613649 | 1:200 |
| Antibody | Anti-mouse CD31 PerCP-Cy5.5 (Rat monoclonal) | Biolegend | Cat#102420 RRID:AB_10613644 | 1:200 |
| Antibody | Anti-mouse PD-L1 BV421 (Rat monoclonal) | Biolegend | Cat#124315 RRID:AB_10897097 | 1:200 |
| Antibody | Anti-mouse CD8a APC-Cy7 (Rat monoclonal) | Biolegend | Cat#100714 RRID:AB_312753 | 1:400 |

*Continued on next page*

*Continued*

| Reagent type (species) or resource | Designation | Source or reference | Identifiers | Additional information |
|---|---|---|---|---|
| Antibody | Anti-mouse CD44 PerCP-Cy5.5 (Rat monoclonal) | Biolegend | Cat# 103032 RRID:AB_2076204 | 1:400 |
| Antibody | Anti-mouse B220/ CD45R BV510 (Rat monoclonal) | Biolegend | Cat# 103248 RRID:AB_2650679 | 1:300 |
| Antibody | Anti-mouse B220/ CD45R PE (Rat monoclonal) | Biolegend | Cat# 103208 RRID:AB_312993 | 1:300 |
| Antibody | Anti-mouse CD11c APC Cy7 (Hamster monoclonal) | Biolegend | Cat#117324 RRID:AB_830649 | 1:400 |
| Antibody | Anti-mouse CD11b PE-Cy7 (Rat monoclonal) | Biolegend | Cat#101216 | 1:300 |
| Peptide, recombinant protein | Streptavidin BV421 | Biolegend | Cat#405226 | 1:1000 |
| Peptide, recombinant protein | Streptavidin AF488 | Thermo Fisher Scientific | Cat#S11223 | 1:1000 |
| Chemical compound, drug | PolyI:C | Invivogen | Cat#Vac-PIC | |
| Chemical compound, drug | Nystatin | Sigma Aldrich | Cat#N4014 | |
| Chemical compound, drug | Violet proliferation dye | BD Biosciences | Cat#562158 | |
| Chemical compound, drug | CFSE | BD Biosciences | Cat#565082 | |
| Strain, strain background (*Mus musculus*) | WT: C57BL/6 | Charles River Labs | C57BL/6 (B6) Mouse Inbred 027 | |
| Strain, strain background (*M. musculus*) | OT1: C57BL/6-Tg (TcraTcrb) 1100Mjb/J | Jackson Labs | JAX: 003831 | |
| Cell line (*M. musculus*) | SVEC4-10 | ATCC | ATCC CRL2181 | |
| Primary cells (*M. musculus*) | mLEC | Cell Biologics | C57-6092 | |

## Mice

5-6 week-old mice were purchased from Charles River or Jackson Laboratory, unless otherwise stated, bred and housed in the University of Colorado Anschutz Medical Campus Animal Barrier Facility. Wild type and OT1 mice were all bred on a C57BL/6 background. OT1 mice are a TCR transgenic strain specific to the SIINFEKL peptide of ova (OVA257-264) in the context of H-2K$^b$. All animal procedures were approved by the Institutional Animal Care and Use Committee at the University of Colorado.

## Phosphorothioate and phosphodiester oligonucleotides

Oligonucleotides were synthesized by Integrated DNA Technologies (IDT) and contained a 5' amine for conjugation, primer binding site, barcode, 10x Genomics Gel Bead Primer binding site for capture sequence 2, and a 3' biotin. Phosphorothioate oligonucleotides contained a phosphorothioate

modification at every linkage. All oligonucleotide sequences can be found in *Figure 1—source data 1*.

## Conjugation of oligonucleotides to protein

Oligonucleotides were conjugated to ova by iEDDA-click chemistry (*van Buggenum et al., 2016*). Oligonucleotides were derivatized with trans-cyclooctene (TCO) in 10× borate buffered saline (BBS; 0.5 M borate, 1.5 M NaCl, pH 7.6; sterile filtered). Dilution of this buffer to 1× results in a final pH of 8.5. A reaction containing 40 nmol of amine-modified oligo (0.5 mM), 1× BBS, 10% DMSO, 8 µL of 100 mM TCO-PEG4-NHS in DMSO (10 mM final; Click Chemistry Tools, A137), pH 8.5 was rotated at room temperature for 15 min. A second aliquot containing the same amount of TCO-PEG4-NHS in DMSO was added, and the reaction was rotated at room temperature for another hour. Excess NHS was quenched by adding glycine, pH 8.5 to a final concentration of 20 mM and rotated at room temperature for 5 min. Modification was confirmed by analysis on an 8% denaturing TBE PAGE gel. Samples were precipitated by splitting the reaction into 20 µL aliquots and adding 280 µL of nuclease-free water, 30 µL of 3 M NaCl, and 990 µL of 100% ethanol. The precipitation reaction was incubated at −80°C overnight, followed by centrifugation at >10,000,000 ×*g* for 30 min. The supernatant was discarded, the pellet was washed with 100 µL of 75% ethanol, and centrifuged at >10,000,000 ×*g* for 10 min. The supernatant was removed, and the pellets were dried for 5 min at room temperature. The pellets were recombined by resuspension in 50 µL of 1× BBS. Samples were quantified by $A_{260}$.

To conjugate methyltetrazine to ova, detoxified ova (Sigma-Aldrich, St. Louis, MO) (using a Triton X-114 lipopolysaccharide detoxification method; *Anis et al., 2007*) was buffer exchanged into 1× BBS, pH 8.5. To an Amicon 0.5 mL 30 kDa filter (Millipore, UFC5030) was added 1 mg of ova and 1× BBS to a volume of 450 µL. The filter was centrifuged at 14,000 ×*g* for 5 min. The flow through was discarded and the sample washed twice with 400 µL of 1× BBS. The product-containing column was inverted into a clean collection tube and centrifuged at 1000 ×*g* for 2 min. Assuming no loss, the volume of the sample was adjusted to 2 mg/mL with 1× BBS. 400 µL of 1× BBS was added to the Amicon filter and stored at 4°C for later use. A 500 µL labeling reaction containing 0.5 mg of ova in 1× BBS and 50 µL of 2 mM mTz-PEG4-NHS in DMSO (0.2 mM final; Click Chemistry Tools, 1069), pH 8.5 was rotated at 4°C overnight. Excess NHS was quenched by adding glycine, pH 8.5 to a final concentration of 20 mM and rotated at room temperature for 10 min. The previously stored Amicon filter was centrifuged at 14,000 ×*g* for 5 min and the flow through discarded. 400 µL of reaction mixture was added to the filter and centrifuged at 14,000 ×*g* for 5 min. This was repeated until all 1 mg of protein had been added to the filter and was supplemented with 1× BBS as needed. Samples were washed 1× with 400 µL of 1× BBS. The product-containing column was inverted into a clean collection tube and centrifuged at 1000 ×*g* for 2 min. Assuming no loss, the volume of the sample was adjusted to 5 mg/mL with 1× BBS.

For the final antigen-DNA conjugation, a 100 µL reaction containing 300 µg of ova-mTz and 6 nmol of oligonucleotide-TCO (1:1 equivalents) in 1× BBS was rotated at 4°C overnight. Excess mTz was quenched with 10 µL of 10 mM TCO-PEG4-glycine and rotated at room temperature for 10 min. TCO-PEG4-glycine was prepared by reaction of 10 mM TCO-PEG4-NHS with 20 mM glycine, pH 8.5 in 1× BBS for 1 hr at room temperature and stored at −20°C. Products were analyzed by 10% TBE PAGE. For purification, *e*xcess ova and DNA were removed by filter centrifugation. 200 µL of 1× PBS was added to an Amicon 0.5 mL 50 kDa filter (Millipore, UFC5050) followed by 300 µL of sample. The filter was centrifuged at 14,000 ×*g* for 5 min and the flow through discarded. Samples were washed five times with 400 µL of 1× PBS and centrifuged at 14,000 ×*g* for 5 min. The product-containing column was inverted into a clean collection tube and centrifuged at 1000 ×*g* for 2 min. Purified products were analyzed by 10% TBE PAGE and total protein quantified with Bio-Rad protein quantification reagent (Bio-Rad, 5000006). LPS contamination after conjugation was below 0.5 EU/mg as mentioned in the 'Vaccinations' section.

## Bone marrow-derived DC, macrophages, and LEC cultures

Both left and right tibia and femur were isolated under sterile conditions. Bone marrow was extracted from femurs of 6–8-week-old C57BL/6 mice by decollating the top and bottom of the bone and releasing the marrow with 27 gauge syringe and 5 mL of Modified Essential Medium

(MEM) (Cellgro). Suspension was strained through 100 µm filter, pressed with the back of a syringe and washed. Cells were spun 1500 RPM, 5 min then suspended in minimum essential medium (MEM) with 10% FBS, 20 ng/mL of Granulocyte-macrophage colony-stimulating factor (GM-CSF) from the supernatant of the B78hi-GM-CSF cell line. Every 2 days, dead cellular debris was spun, supernatant collected and combined 1:1 with new 40 ng/mL GM-CSF 20% FBS (2×) in MEM. After 7 days of culturing at 37℃, 5% CO2 cells were harvested for respective assays. Mouse LECs (Cell Biologics, C57-6092) were cultured in Endothelial Cell Media (Cell Biologics, M1168) with kit supplement. T75 Flasks were coated with gelatin for 30 min 37℃, washed with PBS, and then inoculated with mLEC. Cells were passaged with passive trypsin no more than 3–6 times and split at density of 1:2. SVEC4-10 (ATCC CRL2181), an SV40 transformed endothelial cell line, was purchased from ATCC and mycoplasma tested before use. SVECs have been characterized to be similar to LECs (*Xiong et al., 2017*), and CD31 and PDPN expression were validated prior to use. SVEC were cultured in RPMI with 10% FBS and passaged with passive trypsin and split at a density of 1:3. For BMDMs, whole bone marrow was isolated and red blood cells were lysed. Cells were then cultured in M-CSF (50 ng/mL) for 6 days in complete media. Cells were harvested via cell scraper and plated for treatment.

## Conjugate detection assay

Dendritic cells (BMDC), endothelial cells (mLEC), or SV-40 transformed endothelial cells (SVECs) or BMDM cultures were stimulated with 5 µg of either ova-psDNA or ova with or without 20 µg of anti-CD40, 20 µg Poly I:C in a 6-well format. 24 hr post treatment, cells were washed and refreshed with new media. At designated time points, cells were harvested, counted, and transferred into microcentrifuge tubes, spun at 350 g, and both supernatant and pellets were frozen at −80℃. Cell pellets were lysed in 50 µL of Mammalian Protein Extraction Reagent (Thermo Scientific, 78503). Conjugate DNA was measured by qPCR amplification from 1 µL of lysate in a 10 µL reaction containing 5 µL of iTaq Universal SYBR Green Supermix (Bio-Rad, 1725125) and 5 pmol of each primer (*Figure 1— source data 1*). Quantification was measured using an external standard curve and normalized to lysate protein content. To visualize within ova-psDNA acquisition by cells, cells were fixed with 10% formalin for 10 min at room temperature in the dark, washed with PBS, and spun for 10 min at 2000 rpm. Cells were then permeabilized with 100% ice-cold methanol for 20 min at −20℃. Cells were then washed with PBS and spun as above. Cells were stained with the anti-ova antibody as above for at least 2 hr at room temperature and then washed with 1% bovine serum albumin (BSA) with sodium azide (FACS buffer) and spun as above. Cells were then incubated with an anti-rabbit secondary in PE for 1 hr at room temperature and then washed with FACS buffer. Cells were then stained with streptavidin conjugated to BV421 in PBS for 15 min at room temperature and then washed twice with FACS buffer prior to acquiring cells on a FACS CANTO II flow cytometer. Analysis was performed using FlowJo software. Immunofluorescence was performed as above except cells were grown on glass coverslips and stained on cover slips using an anti-rabbit dylight 649 and streptavidin-FITC. Coverslips were mounted with Vectashield with DAPI and imaged on a Zeiss LSM780 confocal microscope. The imaging experiments were performed in the Advanced Light Microscopy Core part of the NeuroTechnology Center at University of Colorado Anschutz Medical Campus supported in part by the Rocky Mountain Neurological Disorders Core Grant Number P30 NS048154 and by the Diabetes Research Center Grant Number P30 DK116073. Contents are the authors' sole responsibility and do not necessarily represent official NIH views.

## OT1 isolation and co-culture

CD8 T cells were isolated from an OT1+ mouse using the mojosort mouse CD8 T cell isolation kit (Biolegend) and labeled with violet proliferation dye (BD Biosciences cat# 562158). For DC-T cell co-culture, BMDCs were treated with psOVA (5 µg), or ova+psDNA (5 µg) for 1, 3, or 7 days. BMDCs were washed and then co-cultured with labeled OT1s for 3 days at a 1:10 ratio of BMDC:OT1. Cells were then stained and run on a flow cytometer. OT1 division (percent dividing cells) was calculated as previously described (*Roederer, 2011*) using the equation fraction diluted $= \sum_{1}^{i} \frac{N_i}{2^i} / \sum_{0}^{i} \frac{N_i}{2^i}$, where $i$ is the generation number (0 is the undivided population), and $N_i$ is the number of events in generation $i$.

## Vaccinations

6–8-week-old C57BL/6 (CD45.2) mice were immunized with 1E3 or 1E4 colony-forming units (CFU) of Vaccinia Western Reserve or 5 µg of poly I:C (Invivogen) with or without 5 µg of anti-CD40 (FGK4.5, BioXcell)and 10 µg of ova-psDNA or ova in 50 µL volume by footpad injection. Endotoxin levels were quantified using the Pierce Limulus Amebocyte Lysate Chromogenic Endotoxin Quantitation kit (Thermo Scientific) to be less than 0.5 EU/mg for either ova or ova conjugated to psDNA.

## Nystatin

Nystatin (Sigma N4014) was resuspended in DMSO to a concentration of 10 mg/mL. Mice were injected with 50 µL of 10 mg/mL nystatin per footpad 1 hr prior to injection with ova conjugated to Alexa 488 (5 µg) in a mixture with polyI:C and anti-CD40 (2.5 µg each). LNs were harvested and digested as below (preparation of single-cell suspensions) and stained with CD45 brilliant violet 510 (Biolegend clone 30F11, 1:300), PDPN APC (Biolegend clone 8.1.1, 1:200), CD31 PercP Cy5.5 (Biolegend clone 390, 1:200), and PD-L1 pacific blue (Biolegend clone 10F.9G2, 1:200).

## Tetramer and intracellular cytokine assays

Draining LNs were processed by glass slide maceration 7 days after injection, washed, and suspended in FACS (2% FBS in PBS) buffer containing Tetramer (SIINFEKL)-PE (1:400) (NIH tetramer core facility), CD8 APC-Cy7 (Biolegend clone 53-6.7 1:400) for 1 hr at 37C. Cells were washed and stained for 30 min in CD44 PerCP Cy5.5 (Biolegend clone IM7, 1:400), B220 BV510 (Biolegend clone RA3-6B2, 1:300). Samples were ran on the FACS Canto II flow cytometer (BD).

## Preparation of single-cell suspensions

2 days or 2 weeks following vaccination with 1E3 CFU of VV-WR with 10 µg of ova-psDNA per footpad, popliteal LNs were removed from 15 mice and LNs were pulled apart with 22-gauge needles. Tissue was digested with 0.25 mg of Liberase DL (Roche, Indianapolis, IN) per mL of EHAA media with DNAse (Worthington, Lakewood, NJ) at 37°. Every 15 min media was removed, cells spun down, and new digestion media added to the undigested tissue until no tissue remained, ~1 hr. Following digestion, cells were filtered through a screen and washed with 5 mM EDTA in EHAA. LN cells were then divided into thirds where one-third underwent staining with CD11c (N418), CD11b and B220, and a live/dead dye (Tonbo). Live cells were then sorted into four tubes on a FACS Aria Cell Sorter (BD): sorted CD11c-APC Cy7 (Biolegend clone N418 1:400)+ cells, sorted CD11b PE-Cy7 (Biolegend clone M1/70)+ cells, sorted B220 PE (Biolegend clone RA3-6B2)+ cells and Fixable Viability Stain 510 (BD Biosciences Cat # 546406) ungated live cells, which were recombined at a 4:4:1:1 ratio, respectively. For the remaining two-thirds of cells, cells were stained with CD45 PE followed by magnetic bead isolation using the Miltenyi bead isolation kit. CD45-negative cells that passed through the column were then washed. Both sorted and selected (CD45+ and CD45-) cells were then washed with PBS in 0.1% BSA as described in the Cell Prep Guide (10x Genomics) and counted using a hemacytometer. Final concentration of cells was approximately 1000 cells/µL and approximately 10–20 µL were assayed.

## Single-cell library preparation using the 10x Genomics platform

Cells were assayed using the 10x Genomics single-cell 3' expression kit v3 according to the manufacturer's instructions (CG000183 Rev B) and CITE-seq protocol (cite-seq.com/protocol Cite-seq_190213) with the following changes:

1. cDNA amplification and cleanup. During cDNA amplification, 1 µL of 0.2 µM each mixture of additive forward and reverse primers (*Figure 1—source data 1*) was included to amplify the antigen tags. The CITE-seq protocol was followed for size selection and cleanup of the cDNA and antigen tag products. Antigen tag products were eluted in 60 µL of nuclease-free water.
2. Amplification of antigen tag sequencing libraries. A 100 µL PCR reaction was prepared containing 45 µL of purified antigen tag products, 1X Phusion HF Buffer (NEB), 200 µM dNTPs, 25 pmol each Illumina sequencing forward and reverse primers (*Figure 1—source data 1*), 2 Units Phusion High Fidelity DNA Polymerase. PCR cycling conditions were 95°C for 3 min, 6-10× (95°C for 20 s, 60°C for 30 s, 72°C for 20 s), 72°C for 5 min. Products were purified according to the CITE-seq protocol. Gene expression and antigen tag libraries were analyzed on the Agilent

D1000 Tapestation and quantified using the Qubit HS dsDNA fluorometric quantitation kit (Thermo Scientific).

All libraries were sequenced on a Illumina NovaSeq 6000 with 2 × 150 base pair read lengths.

## Transcriptome and oligonucleotide detection and analysis

Briefly, FASTQ files from the gene expression and antigen tracking libraries were processed using the feature barcode version of the cellranger count pipeline (v3.1.0). Reads were aligned to the mm10 and vaccinia virus (NC_006998) reference genomes. Analysis of gene expression and antigen tracking data was performed using the Seurat R package (v3.2). Antigen tracking and gene expression data were combined into the same Seurat object for each sample (CD45-/day 2, CD45+/day 2, CD45-/day 14, CD45+/day 14). Cells were filtered based on the number of detected genes (>250 and <5000) and the percent of mitochondrial reads (<15%). Gene expression counts were log-normalized (NormalizeData), and relative ova signal was calculated by dividing ova-psDNA counts by the median ova-psDNA counts for all T and B cells present in the sample. To allow for the values to be log-transformed for visualization, a pseudo count was added (smallest non-zero value * 0.5).

Gene expression data were scaled and centered (ScaleData). 2000 variable features (FindVariableFeatures) were used for PCA (RunPCA), and the first 40 principal components were used to find clusters (FindNeighbors, FindClusters) and calculate uniform manifold approximation and projection (UMAP) (RunUMAP). Cell types were annotated using the R package clustifyr (https://rnabioco.github.io/clustifyr) (*Fu et al., 2020*) along with reference bulk RNA-seq data from ImmGen (available for download through the clustifyrdata R package, https://rnabioco.github.io/clustifyrdata). To annotate cell subtypes, the samples were divided into separate objects for DCs, LECs, and FRCs and reprocessed (FindVariableFeatures, ScaleData, RunPCA, RunUMAP, FindNeighbors, FindClusters). Cell subsets were annotated using clustifyr with reference bulk RNA-seq data for DCs (*Brown et al., 2019*; *Miller et al., 2012*), FRCs (*Rodda et al., 2018*), and LECs (*Fujimoto et al., 2020*; *Kalucka et al., 2020*; *Xiang et al., 2020*). After assigning DC, LEC, and FRC subtypes, the other cell types (T/B cells, epithelial cells, NK cells) were added back to the objects and reprocessed as described above.

Identification of ova-low and -high populations was accomplished using a two-component Gaussian mixture model implemented with the R package mixtools (https://cran.r-project.org/web/packages/mixtools/index.html). All LECs were used when identifying ova-low and ova-high cells (*Figure 4*). For DCs (*Figure 3—figure supplement 5*), ova-low and -high populations were identified independently for each DC cell type. For ova-low and ova-high populations, differentially expressed genes were identified using the R package presto (wilcoxauc, https://github.com/immunogenomics/presto). Differentially expressed genes were filtered to include those with an adjusted p-value<0.05, log fold-change > 0.25, area under the receiver operator curve (AUC) > 0.5, and with at least 50% of ova-high cells expressing the gene.

## Raw data and analysis software

Raw and processed data for this study have been deposited at NCBI GEO under accession GSE150719. A reproducible analysis pipeline is available at https://github.com/rnabioco/antigen-tracking http://doi.org/10.5281/zenodo.4615724 (*Sheridan and Hesselberth, 2021*; copy archived at swh:1:rev:f7f6c0696f08aeeac6ad88c39975197a0791e30d).

## Statistical analysis

Statistical analysis was done using either a non-parametric two-tailed Mann–Whitney t-test or multiple t-tests with a two-stage step-up method of Benjamini, Krieger, and Yekutieli without assuming consistent standard deviations. A biological replicate was considered a measurement of a biologically distinct sample (such as a separate mouse), and a technical replicate was considered a repeated measurement of the same sample. Each in vivo analysis was performed with 3–6 mice per group as determined by a power calculation using the assumption (based on prior data) that there will be at least a twofold change with a standard deviation of less than 0.5. To calculate numbers, we performed a power calculation with an α of 0.5 and a 1-β of 0.80 to determine at least three mice per group are evaluated. Error bars indicate the standard error of the mean (SEM), and all analyses were blinded.

## Acknowledgements

The K$^b$ SIINFEKL PE tetramer was provided by the NIH Tetramer Core Facility. We thank the HIMSR flow cytometry core facility for use of the Aria cell sorter and University of Colorado Anschutz Medical Campus Genomics Core Facility (NIH P30 CA046934).

## Additional information

### Funding

| Funder | Grant reference number | Author |
|---|---|---|
| National Institutes of Health | R01 AI121209 | Beth Ann Jiron Tamburini |
| University of Colorado Denver | Outstanding Early Career Scholar and RBI Clinical Scholar Award | Beth Ann Jiron Tamburini |
| American Cancer Society | Post-doctoral Fellowship | Shannon M Walsh |
| National Institutes of Health | T32 AI007405 | Erin D Lucas |
| National Institutes of Health | R35 GM119550 | Jay R Hesselberth |
| National Institutes of Health | T32 AI074491 | Ryan M Sheridan |
| National Institutes of Health | R21 AI155929 | Jay R Hesselberth Beth Ann Jiron Tamburini |

The funders had no role in study design, data collection and interpretation, or the decision to submit the work for publication.

### Author contributions

Shannon M Walsh, Conceptualization, Resources, Formal analysis, Validation, Methodology, Writing - review and editing; Ryan M Sheridan, Resources, Data curation, Software, Formal analysis, Validation, Investigation, Visualization, Methodology, Writing - review and editing; Erin D Lucas, Conceptualization, Data curation, Formal analysis, Validation, Investigation, Visualization, Methodology, Writing - review and editing; Thu A Doan, Data curation, Formal analysis, Investigation, Writing - review and editing; Brian C Ware, Data curation, Formal analysis, Investigation, Visualization, Writing - review and editing; Johnathon Schafer, Validation, Investigation, Methodology, We have included Mr. Johnathon Schafer as an additional author that contributed to the revisions of the paper. He provided assistance with the revised figures and assisted with the design and interpretation of the experiments. All authors are aware and accepting of his addition to the author list; Rui Fu, Resources, Methodology; Matthew A Burchill, Formal analysis, Investigation, Writing - review and editing; Jay R Hesselberth, Conceptualization, Resources, Data curation, Formal analysis, Supervision, Validation, Investigation, Visualization, Methodology, Writing - review and editing; Beth Ann Jiron Tamburini, Conceptualization, Resources, Data curation, Formal analysis, Supervision, Funding acquisition, Validation, Investigation, Visualization, Methodology, Writing - original draft, Project administration, Writing - review and editing

### Author ORCIDs

Shannon M Walsh ![iD] https://orcid.org/0000-0001-9845-629X
Jay R Hesselberth ![iD] https://orcid.org/0000-0002-6299-179X
Beth Ann Jiron Tamburini ![iD] https://orcid.org/0000-0003-1991-231X

### Ethics

Animal experimentation: All animal procedures were approved by the Institutional Animal Care and Use Committee at the University of Colorado under protocol number 00067.

### Decision letter and Author response

Decision letter https://doi.org/10.7554/eLife.62781.sa1

Author response https://doi.org/10.7554/eLife.62781.sa2

## Additional files

### Supplementary files

• Transparent reporting form

### Data availability

Raw and processed data for this study have been deposited at NCBI GEO under accession GSE150719. A reproducible analysis pipeline is available at https://github.com/rnabioco/antigen-tracking andhttps://zenodo.org/record/4615724 (copy archived at https://archive.softwareheritage.org/swh:1:rev:f7f6c0696f08aeeac6ad88c39975197a0791e30d) .

The following dataset was generated:

| Author(s) | Year | Dataset title | Dataset URL | Database and Identifier |
|---|---|---|---|---|
| Walsh SM, Sheridan RM, Lucas ED, Doan TA, Ware BC, Schafer J, Fu R, Burchill MA, Hesselberth JR, Tamburini BAJ | 2020 | Molecular tracking devices quantify antigen distribution and archiving in the lymph node | https://www.ncbi.nlm.nih.gov/geo/query/acc.cgi?acc=GSE150719 | NCBI Gene Expression Omnibus, GSE150719 |

The following previously published datasets were used:

| Author(s) | Year | Dataset title | Dataset URL | Database and Identifier |
|---|---|---|---|---|
| Brown CC, Gudjonson H, Pritykin Y, Deep D, Lavallee V, Mendoza A, Fromme R, Mazutis L, Ariyan C, Leslie C, Pe'er D, Rudensky AY | 2019 | Transcriptional basis of mouse and human dendritic cell heterogeneity revealed by single-cell profiling | https://www.ncbi.nlm.nih.gov/geo/query/acc.cgi?acc=GSE137710 | NCBI Gene Expression Omnibus, GSE137710 |
| Aguilar SV, Aguilar O, Allan R, Amir ADD, Angeli V, Artyomov MN, Asinovski N, Astarita J, Austen F, Bajpai G, Barrett N, Baysoy A, Benoist C, Bellemare-Pelletier A, Berg B, Best A, Bezman N, Blair D, Blander JM, Bogunovic M, Brennan P, Brenner M, Brown B, Buechler M, Buenrostro J, Casanova MA, Choi K, Chow A, Chudnovskiy A, Cipoletta D, Cohen N, Collins JJ, Colonna M, Cook A, Costello J, Cremasco V, CrowlT, Crozat K, Cruse R, D'Angelo J, Dalod M, Davis | 2009 | ImmGen Microarray Phase 1 | https://www.ncbi.nlm.nih.gov/geo/query/acc.cgi?acc=GSE15907 | NCBI Gene Expression Omnibus, GSE15907 |

S, Demiralp C,
Deng T, Desai JV,
Desland F,
Dhainaut M, Ding J,
Doedens A,
Dominguez C,
Doran G, Dress R,
Dustin M, Dwyer D,
Dzhagalov I, Elpek
K, Ergun A, Ericson
J, Esomonu E,
Fairfax K, Fletcher
A, Frascoli M,
Fuchs A, Gainullina
A, Gal-Oz S,
Gallagher M,
Gautier E, Gazit R,
Gibbings S, Giraud
M, Ginhoux F,
Goldrath A,
Gotthardt D, Gray
D, Greter M,
Grieshaber-Bouyer
R, Guilliams M,
Haidermota S,
Hardy R,
Hashimoto D, Helft
J, Hendricks D,
Heng T, Hill J,
Hyatt G, Idoyaga J,
Jakubzick C,
Jarjoura J, Jepson
D, Jia B, Jianu R,
Johanson T, Jordan
S, Jojic V,
Kamimura Y, Kana
V, Kang J, Kapoor
V, Kenigsberg E,
Kent A, Kim C, Kim
E, Kim F, Kim J,
Kim K, Kiner E,
Knell J, Koller D,
Kozinn L, Krchma K,
Kreslavsky T,
Kronenberg M,
Kwan W-H, Laidlaw
D, Lam V, Lanier L,
Laplace C, Lareau
C, Lavin Y, Lavine
KJ, Leader A,
Leboeuf M, Lee J,
Li B, Li H, Li Y,
Lionakis MS, Luche
H, Lynch L, Magen
A, Maier B,
Malhotra D,
Malhotra N,
Malissen M,
Maslova A, Mathis
D, McFarland A,
Merad M, Meunier
E, Miller J, Milner
J, Mingueneau M,
Min-Oo G, Monach
P, Moodley D,
Mortha A, Morvan
M, Mostafavi S,
Muller S, Muus C,
Nabekura T, Rao
TN, Narang V,
Narayan K, Ner-
Gaon H, Nguyen Q,
Nigrovic PA,
Novakovsky G, Nutt

S, Omilusik K, Ortiz-Lopez A, Paidassi H, Paik H, Painter M, Paynich M, Peng V, Potempa M, Pradhan R, Price J, Qi Y, Quon S, Ramirez R, Ramanan D, Randolph G, Regev A, Rhoads A, Robinette M, Rose S, Rossi D, Rothamel K, Sachidanandam R, Sathe P, Scott C, Seddu K, See P, Sergushichev A, Shaw L, Shay T, Shemesh A, Shinton S, Shyer J, Sieweke M, Smillie C, Spel L, Spidale N, Stifano G, Subramanian A, Sun J, Sylvia K, Tellier J, This S, Tomasello E, Todorov H, Turley S, Vijaykumar B, Wagers A, Wakamatsu E, Wang C, Wang PL, Wroblewska A, Wu J, Yang E, Yang L, Yim A, Yng LS, Yoshida H, Yu B, Zhou Y, Zhu Y, Ziemkiewicz C, The Immunological Genome Consortium

| | | | | |
|---|---|---|---|---|
| Rodda LB, Cyster JG | 2018 | Single-cell RNA sequencing of lymph node stromal cells reveals niche-associated heterogeneity | https://www.ncbi.nlm.nih.gov/geo/query/acc.cgi?acc=GSE112903 | NCBI Gene Expression Omnibus, GSE112903 |
| Xiang M, Ulvmar MH, Butcher EC, Brulois K, Nordling S | 2020 | Single-cell RNA-seq of the mouse lymph node lymphatic vasculature: Droplet-seq | https://www.ncbi.nlm.nih.gov/geo/query/acc.cgi?acc=GSE145121 | NCBI Gene Expression Omnibus, GSE145121 |
| Yoshida H, Lareau CA, Ramirez RN, Rose SA | 2018 | ImmGen ULI: Systemwide RNA-seq profiles (#1) | https://www.ncbi.nlm.nih.gov/geo/query/acc.cgi?acc=GSE109125 | NCBI Gene Expression Omnibus, GSE109129 |

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
