## [Decision Letter]

**Acceptance summary:**

You have developed a powerful method to track in vivo the cellular distribution of an immunogenic protein-DNA complex overlaid with the rich information from single cell RNA seq. This has enabled you to quantify the archiving of antigens in lymphatic endothelial cells, better understand the molecule pathways involved in archiving, and put this in context with the distribution of the protein-DNA complexes among dendritic populations at early and late time points. The work will be of interest to immunologists and vascular biologists, and the general approach may find application in tracking the fate of macromolecules incorporating oligonucleotides in other contexts.

**Decision letter after peer review:**

Thank you for submitting your article "Molecular tracking devices quantify antigen distribution and archiving in the lymph node" for consideration by *eLife*. Your article has been reviewed by 3 peer reviewers, including Michael L Dustin as the Reviewing Editor and Reviewer #1, and the evaluation has been overseen by Carla Rothlin as the Senior Editor.

The reviewers have discussed the reviews with one another and the Reviewing Editor has drafted this decision to help you prepare a revised submission.

Summary:

The idea of using DNA tags to enable tracking of protein and its subsequent handling is innovative and interesting. The manuscript is well written and some of the notions presented may help drive the field forward. There is a potential to revise the manuscript for *eLife* as a proof of principle for the approach of using scRNA seq to track an antigen in vivo.

Essential revisions:

While the archiving of the psDNA-Ova conjugate in lymphatic endothelial cells matches the earlier observations of fluorescently tagged Ova, the power of this method is to work with scRNAseq to be able to identify novel cells that interact the complex. In this regard, it's clear that the psDNA-Ova may interact differently with various cell types due to the potential for recognition of the psDNA, particularly by TLR9 as suggested. Tracking the psDNA-Ova conjugate as an immunogenic adjuvant-antigen complex is an interesting starting point. You can address this concern by reframing the goal from tracking a protein antigen to characterizing the archiving and presentation of the barcoded psDNA-Ova complex. This doesn't require any additional work, just changing the way you set it up- that this if you immunogen is a DNA-protein complex- you can track it by scRNA-seq. The potential to study trafficking in a TLR9 KO mice in the future might open up using this method more generically for tracking the protein antigen, but this would require much more work to rule out other influence of the DNA on archiving and processing of the antigen.

Your evidence that the psDNA really reflects the distribution of the protein is not sufficient. In Figure 1 c and d it's not clear how you measure the amount of protein? Is this the protein injected or the protein detected in a immunoblot or capture immunoassay? To extend the in vitro analysis in Figure 1c and d to actually visualize the native protein with anti-Ova and the psDNA by FISH would provide a high degree of clarity that the psDNA is not acting like a tattoo that outlives the intact protein. The FISH method could take advantage of any amplification step as long as it is consistent with detection by scRNAseq. Microscopy could demonstrate that the two signals remain in the same comparments. A particularly powerful way to show the direct association would be to perform a bulk IP-seq with anti-Ova and detection of the bar code with a test of the efficiency of depletion of the bar code form the cell lysate. A well-controlled experiment could be performed to map out the time dependent loss of protein (IP-western or capture immunoassay), and the free and protein associated psDNA. It would be ideal to include a macrophage in the analysis as a highly degradative cells in comparison to the dendritic cell and LEC, that maybe more specialized to regain intact proteins.

---

## [Author Response]

Essential revisions:While the archiving of the psDNA-Ova conjugate in lymphatic endothelial cells matches the earlier observations of fluorescently tagged Ova, the power of this method is to work with scRNAseq to be able to identify novel cells that interact the complex. In this regard, it's clear that the psDNA-Ova may interact differently with various cell types due to the potential for recognition of the psDNA, particularly by TLR9 as suggested. Tracking the psDNA-Ova conjugate as an immunogenic adjuvant-antigen complex is an interesting starting point. You can address this concern by reframing the goal from tracking a protein antigen to characterizing the archiving and presentation of the barcoded psDNA-Ova complex. This doesn't require any additional work, just changing the way you set it up- that this if you immunogen is a DNA-protein complex- you can track it by scRNA-seq. The potential to study trafficking in a TLR9 KO mice in the future might open up using this method more generically for tracking the protein antigen, but this would require much more work to rule out other influence of the DNA on archiving and processing of the antigen.

Thank you for this suggestion, we edited the text to clearly indicate that we are characterizing the archiving and presentation of a barcoded psDNA-ova complex as a new vaccination method that has unique features. We have also highlighted that antigen archiving that occurs following combined TLR/anti-CD40 has similar characteristics to ova-psDNA vaccination. Our future studies are aimed at identifying differences and using the TLR9KO mice to more generically track the protein antigen, as mentioned by the reviewer.

Your evidence that the psDNA really reflects the distribution of the protein is not sufficient. In Figure 1 c and d it's not clear how you measure the amount of protein?

We added text to better explain how we quantified the amount of DNA. Lines 84-86. We have added additional text and figures to demonstrate that the amount of protein correlates with the amount of DNA present. New figure 1d,e and supplementary figure 2a,b,f,g. Lines 89-93, 96-98, 101-105

Is this the protein injected or the protein detected in a immunoblot or capture immunoassay?

In the figure you are referring to the amount of protein detected is the total protein within the cell to normalize the amount of DNA to variations in the lysis of the cells over time and across experiments. The question I believe you are asking is how we know how much of the ova protein is detected in the cell when conjugated to the DNA and what is the relative amount. As stated above we have performed several new experiments to demonstrate that we can detect ova and psDNA within the same cells and co-localized together in addition to demonstrating ova presentation by the dendritic cells in figure 1. These new experiments and text can be found lines 89-105 and figures 1d,e and figure 1—figure supplement 1a,b,c,g

To extend the in vitro analysis in Figure 1c and d to actually visualize the native protein with anti-Ova and the psDNA by FISH would provide a high degree of clarity that the psDNA is not acting like a tattoo that outlives the intact protein. The FISH method could take advantage of any amplification step as long as it is consistent with detection by scRNAseq. Microscopy could demonstrate that the two signals remain in the same comparments. A particularly powerful way to show the direct association would be to perform a bulk IP-seq with anti-Ova and detection of the bar code with a test of the efficiency of depletion of the bar code form the cell lysate. A well-controlled experiment could be performed to map out the time dependent loss of protein (IP-western or capture immunoassay), and the free and protein associated psDNA.

We agree that the methods described above would demonstrate the degree of conjugation. In the creation of our oligo we not only changed the phosphodiester bonds to the phosphorothioate bonds, but also included a 3’ biotin modification (Figure 1a and Figure 1-source data 1). As the 5’ end of the oligo was covalently linked to the protein this allowed us to use the 3’ biotin for detection of the oligo. If the oligo was degraded the 3’ biotin would be lost. We used the biotinylation of the DNA tag to validate the conjugation of ova to the psDNA as an alternative to FISH. We detected both ova and psDNA within the same location of cells (Figures 1e and Figure 1—figure supplement 1c) and within the same cells using flow cytometry (Figure 1d and Figure 1—figure supplement 1). Finally, we demonstrated that the protein and the DNA (via detection of the biotin) are lost from BMDCS over the 7 day time period (Figure 1d) in line with what we show in (Figure 1g-loss of antigen presentation). We hope the reviewer will agree that these assays, while not all of the exact methodology suggested above, address the questions raised.

It would be ideal to include a macrophage in the analysis as a highly degradative cells in comparison to the dendritic cell and LEC, that maybe more specialized to regain intact proteins.

We appreciate this comment and included in vitro analysis of bone marrow derived macrophages (BMDM)s and in Figure 1—figure supplement 1g to supplement our BMDC and LEC data. We do indeed find that in vitro cultured macrophages take up a significant amounts of ova-psDNA and process the psDNA more efficiently than the ovalbumin. We hypothesize this is due to higher levels of DNAse II within the lysosomes of macrophages required for the degradation of chromosomal DNA found within apoptotic cells (PMID: 12181746,17979851). We have included this in the text lines:101-105.